# Estimating Noise Transition Matrix with Label Correlations for Noisy Multi-Label Learning

**Shikun Li**[1,2]    **Xiaobo Xia**[3]    **Hansong Zhang**[1,2]    **Yibing Zhan**[4]

**Shiming Ge**[1,2*]    **Tongliang Liu**[3]

[1] Institute of Information Engineering, Chinese Academy of Sciences

[2] School of Cyber Security, University of Chinese Academy of Sciences

[3] Trustworthy Machine Learning Lab, The University of Sydney  [4] JD Explore Academy

## Abstract

In label-noise learning, the noise *transition matrix*, bridging the class posterior for noisy and clean data, has been widely exploited to learn *statistically consistent* classifiers. The effectiveness of these algorithms relies heavily on estimating the transition matrix. Recently, the problem of label-noise learning in multi-label classification has received increasing attention, and these consistent algorithms can be applied in multi-label cases. However, the estimation of transition matrices in noisy multi-label learning has not been studied and remains challenging, since most of the existing estimators in noisy multi-class learning depend on the existence of anchor points and the accurate fitting of noisy class posterior. To address this problem, in this paper, we first study the *identifiability problem* of the class-dependent transition matrix in noisy multi-label learning, and then inspired by the identifiability results, we propose a new estimator by exploiting label correlations without neither anchor points nor accurate fitting of noisy class posterior. Specifically, we estimate the occurrence probability of two noisy labels to get noisy label correlations. Then, we perform *sample selection* to further extract information that implies clean label correlations, which is used to estimate the occurrence probability of one noisy label when a certain clean label appears. By utilizing the mismatch of label correlations implied in these occurrence probabilities, the transition matrix is *identifiable*, and can then be acquired by solving a simple bilinear decomposition problem. Empirical results demonstrate the effectiveness of our estimator to estimate the transition matrix with label correlations, leading to better classification performance. Source codes are available at https://github.com/tmllab/Multi-Label-T.

## 1  Introduction

In real-world scenarios, an instance is naturally associated with multiple labels, and these labels have *complex entangled correlations* [7]. Recently, the problem of label-noise learning in multi-label classification has received more and more attention [29, 34, 54, 50, 46, 47], since it is time-consuming and expensive to collect large-scale accurate labels and the noisy labels are much cheaper and easier to acquire. In the setting of *noisy multi-label learning*, the multiple labels assigned to an instance may be corrupted simultaneously. That is to say, any label for each class can be flipped with its respective *transition matrix* that denotes the transition relationship from clean labels to noisy labels.

Transition matrix has been utilized to build a series of *statistically consistent* algorithms for *noisy multi-class learning* [32, 52, 48, 8]. The main advantage of these consistent algorithms is that they

---

*Shiming Ge is the corresponding author. (Email: geshiming@iie.ac.cn)

can guarantee to vanish the differences between the classifiers learned from noisy data and the optimal ones from clean data by increasing the size of noisy examples [28, 33, 52, 39].

Fortunately, these statistically consistent algorithms for noisy multi-class learning can also be applied in such noisy multi-label learning with a little modification [54] (more details can be found in Appendix B). However, the effectiveness of these algorithms heavily relies on estimating the transition matrix. Although the estimation of the transition matrix has been investigated in noisy multi-class learning, the estimation of the transition matrix in noisy multi-label learning has not been studied and remains challenging. Specifically, a series of methods [28, 33, 57, 52, 26] has been proposed to estimate the transition matrix for noisy multi-class learning. Most of them assume the existence of anchor points [28, 33, 57] that are defined as the training examples belonging to a particular clean class surely. Nevertheless, the assumption is strong and hard to check when we only have noisy data [52]. Also, the methods need to accurately fit the noisy or intermediate class posterior of anchor points, which are rather difficult in multi-label cases, due to severe positive-negative imbalance [36].

In this paper, to address the problem of estimating the transition matrix in noisy multi-label learning, we consider utilizing label correlations among noisy multiple labels. Specifically, some label correlations that should *not exist* in practice are included in noisy multi-label learning. For example, "fish" and "water" always co-occur, while "bird" and "sky" always co-occur. But, due to label errors, there is a *slight correlation* between "fish" and "sky", which is impractical. At a high level, we can utilize *the mismatch of label correlations* to identify the transition matrix without neither anchor points nor accurate fitting of noisy class posterior.

In more detail, we first focus on the *identifiability problem* of the class-dependent transition matrix in noisy multi-label learning. Accordingly, a new method that estimates the transition matrix by exploiting label correlations is proposed. That is, motivated by the identifiability result that the label correlation of two noisy labels can not suffice to identify the transition matrix in noisy multi-label learning, we utilize *sample selection* to extract useful information from noisy data, which implies clean label correlations to achieve the identifiability. Afterward, we not only estimate the *occurrence probability* of two noisy labels in noisy data, but also of one noisy label when a certain clean label appears in selected data. By utilizing the mismatch of label correlations implied in these occurrence probabilities, we can prove the identifiability, and transform the problem of estimating the transition matrix using label correlations into a problem of *bilinear decomposition*. Finally, with easy frequency counting, we can get a good estimation of the noise transition matrix.

Empirical results illustrate the effectiveness of the proposed estimator for estimating the transition matrix in noisy multi-label learning, and the consistent algorithms with our estimator can achieve better classification performance.

The rest of the paper is organized as follows. In Section 2, we briefly present the problem setting of label-noise learning in multi-label classification. In Section 3, we discuss the identifiability of the transition matrix under such a noisy multi-label setting, and introduce our estimation method. Experimental results are provided in Section 4. The limitations of this work are discussed in Section 5. Finally, we conclude the paper in Section 6.

## 2 Problem Setting

In this section, we introduce the problem setting of label-noise learning in multi-label classification. In what follows, scalars are in lowercase letters, vectors are in lowercase boldface letters, and matrices/variables are in uppercase letters. For simplicity, let $[q] = \{1, \ldots, q\}$.

**Preliminaries.** Let $D$ be the distribution of a pair of random variables $(\boldsymbol{X}, \boldsymbol{Y})$, where $\boldsymbol{X} \in \mathcal{X} \subseteq \mathbb{R}^d$ denotes the variable of instances, and $\boldsymbol{Y} = \{Y^1, Y^2, ..., Y^q\} \in \{0, 1\}^q$ denotes the variable of targets with $q$ possible class labels. As for $\boldsymbol{Y}$, $Y^j = 1$ indicates that the instance $X$ is associated with the class $j$; $Y^j = 0$, otherwise. In multi-label learning, the goal is to learn a function from $D$ which maps each unseen instance $\boldsymbol{x} \in \mathcal{X}$ to proper labels $\boldsymbol{y}$. However, as discussed, $\boldsymbol{Y}$ is hard to be annotated precisely. Before being observed, their true labels are independently flipped and what we can obtain are noisy training examples $\mathcal{D}_t = \{(\boldsymbol{x}_i, \bar{\boldsymbol{y}}_i)\}_{i=1}^n$, where $\bar{\boldsymbol{y}}_i$ denotes noisy labels. Let $\bar{D}$ be the distribution of the noisy random variables $(\boldsymbol{X}, \bar{\boldsymbol{Y}}) \in \mathcal{X} \times \{0, 1\}^q$. In noisy multi-label learning, our goal is to infer proper labels for each unseen instance by *only* using the noisy training examples.

**Noise transition matrix.** The random variables $\bar{Y}^j$ and $Y^j$ for the class $j$ are related through a noise transition matrix $\boldsymbol{T}^j \in [0, 1]^{2 \times 2}$, $j \in [q]$. Generally, the transition matrix depends on instances,

i.e., $T_{ik}^j(\boldsymbol{x}) = P(\bar{Y}^j = k \mid Y^j = i, \boldsymbol{X} = \boldsymbol{x})$. Nevertheless, given only noisy examples, the instance-dependent transition matrix is *non-identifiable* without any additional assumption [52, 56]. For example, both $P(\bar{Y}^j = k|\boldsymbol{X} = \boldsymbol{x}) = \sum_{i=0}^1 T_{ik}^j P(Y^j = i|\boldsymbol{X} = \boldsymbol{x})$ and $P(\bar{Y}^j = k|\boldsymbol{X} = \boldsymbol{x}) = \sum_{i=0}^1 T_{ik}'^j P'(Y^j = i|\boldsymbol{X} = \boldsymbol{x})$ are valid, when $T_{ik}'^j(\boldsymbol{X} = \boldsymbol{x}) = T_{ik}^j(\boldsymbol{X} = \boldsymbol{x})P(Y^j = i|\boldsymbol{X} = \boldsymbol{x})/P'(\bar{Y}^j = i|\boldsymbol{X} = \boldsymbol{x})$. Therefore, in this paper, we assume that the transition matrix is class-dependent and instance-independent [33], i.e., $P(\bar{Y}^j = k \mid Y^j = i, \boldsymbol{X} = \boldsymbol{x}) = P(\bar{Y}^j = k \mid Y^j = i) = T_{ik}^j$. The definition of the class-dependent label noise can be found in Appendix A, where we further discuss its differences with the class-dependent label noise in multi-class cases.

**Consistent algorithms.** The transition matrix bridges the class posterior probabilities for noisy and clean data, i.e., $P(\bar{Y} = k \mid \boldsymbol{X} = \boldsymbol{x}) = \sum_{i=0}^1 T_{ik} P(Y = i \mid \boldsymbol{X} = \boldsymbol{x})$. Thus, it has been exploited to achieve many statistically consistent algorithms in noisy multi-class learning. Specifically, it has been utilized to build risk-consistent estimators via correcting loss functions [28, 33, 52], and to design classifier-consistent estimators via limiting hypotheses, e.g., [33, 9, 62]. Since the multi-label task can be decomposed into multiple conditionally independent binary classification problems, we also can apply these consistent methods in noisy multi-label learning [54]. In this paper, without loss of generality, we adopt a risk-consistent algorithm, i.e., Reweight [28, 52], to learn statistically consistent classifiers with estimated transition matrices. More details can be found in Appendix B.

**Transition matrix estimation.** As inaccurate transition matrices will degenerate the performances of these consistent algorithms, a series of estimation methods [28, 51, 57, 63, 26] have been proposed for noisy multi-class learning to efficiently identify the transition matrix. However, most of them require the assumption of anchor points [33, 57, 26], which is strong and hard to check in multi-label cases when only noisy data are provided [52]. Besides, severe positive-negative imbalance in multi-label learning [36] will make it difficult to accurately approximate the noisy or intermediate class posterior of anchor points, which is crucial for these methods. This motivates us to seek for a better estimator that can do without anchor points and avoid estimating noisy posterior in noisy multi-label learning.

# 3 Estimating Transition Matrices using Label Correlations

In this section, we first study the identifiability problem [30] of class-dependent transition matrices in multi-label cases. Furthermore, inspired by these results, we propose a new estimator to approximate the transition matrix by utilizing label correlations. It is worth pointing out that our estimator demands neither the existence of anchor points nor accurate fitting of noisy class posterior.

## 3.1 Identifiability of transition matrix

Recently, Liu et al. [30] built identifiability of the noise transition matrix on the Kruskal's identifiability results. Inspired by them, with the complex correlations among class labels, we can get some identifiability results of the class-dependent transition matrix in noisy multi-label learning.

Following [30], to define identifiability, we denote an observation space by $\Omega$. For a general parametric space $\Theta$, denote the distribution (probability density function) induced by the parameter $\theta \in \Theta$ on the observation space $\Omega$ as $P_\theta$ [2]. The identifiability for a general parametric space is defined as follows:

**Definition 1** (Identifiability [30]). *The parameter $\theta$ is identifiable if $P_\theta \neq P_{\theta'}, \forall \theta \neq \theta'$.*

For the class-dependent transition matrix $\boldsymbol{T}^j$ for class $j$, the identifiability can be defined as the following, when $\boldsymbol{T}^j$ is a part of $\theta$.

**Definition 2** (Identifiability of $\boldsymbol{T}^j$). *$\boldsymbol{T}^j$ is identifiable if $P_\theta \neq P_{\theta'}$ for $\theta \neq \theta'$, up to the label permutation of class $j$.*

Here, the label permutation of class $j$ means swapping 1 and 0 class values in class $j$, and the rows of $\boldsymbol{T}^j$ will also swap. Note that $\Omega$ does not necessarily include all observable variables. We use an example to better understand it. For example, let $\Omega := \{\bar{Y}^j, \boldsymbol{X}\}, \theta := \{T^j, P(Y^j|\boldsymbol{X})\}$, and $P_\theta := P(\bar{Y}^j \mid \boldsymbol{X})$, the identifiability of $T^j$ can be achieved with the anchor point assumption [33, 57, 26].

To use notations without confusion, for the identifiability of $T^j$ using label correlations, we let $\Omega := \{\bar{Y}^j, \bar{\boldsymbol{Y}}^{-j}\}$, where $-j$ means other classes having correlations with class $j$, $\theta := \{\boldsymbol{T}^j, P(Y^j), P(\bar{\boldsymbol{Y}}^{-j}|Y^j)\}$, and $P_\theta := P(\bar{Y}^j, \bar{\boldsymbol{Y}}^{-j})$. Also, $\Omega$ does not necessarily include all noisy class labels. We need $\Omega$ to provide useful information to achieve the identifiability. The exploration of an effective $\Omega$ and necessary conditions is one of the focuses of the paper.

**Assumption 1.** $P(\bar{Y}^j = 0 \mid Y^j = 1) + P(\bar{Y}^j = 1 \mid Y^j = 0) < 1, j \in [q]$.

Assumption 1 means that the noisy label is agreed with the clean label on average, which is a standard condition for analysis under the class-dependent transition matrix [32, 31].

**Assumption 2.** $P(Y^i = 0 \mid Y^j = 0) \neq P(Y^i = 0 \mid Y^j = 1)$, $i, j \in [q]$ and $i \neq j$.

Assumption 2 means that the multiple labels have correlations between each other, which is satisfied by the most of $(i, j)$ pairs in the real-world dataset (see Appendix C). When considering multi-label learning, the simplest case is having two class labels. In this case, the following theoretical results can be obtained.

**Theorem 1.** *Two noisy labels $\{\bar{Y}^j, \bar{Y}^i\}$ will not suffice to identify $\boldsymbol{T}^j$.*

This result tells us that the label correlations of two noisy labels can not offer enough information to achieve the identifiability of $\boldsymbol{T}^j$. We provide Theorem 2 based on the Kruskal's identifiability result [21, 40].

**Theorem 2.** *If $\bar{Y}^i$ and $\bar{Y}^k$ are independent given $Y^j$, three noisy labels $\{\bar{Y}^j, \bar{Y}^i, \bar{Y}^k\}$ are sufficient to identify $\boldsymbol{T}^j$.*

The assumption that $\bar{Y}^i$ and $\bar{Y}^k$ are independent given $Y^j$ can be satisfied in certain cases, e.g., the occurrences of "blue" and "dolphin" may be independent given "sea" appearing or not. Nevertheless, due to the complex correlations among labels, this assumption is hard to hold in most cases. When the assumption can not hold, these label correlations are no more sufficient to determine $\boldsymbol{T}^j$, as shown in Theorem 3 in the following.

**Theorem 3.** *If $\bar{Y}^i$ and $\bar{Y}^k$ are not independent given $Y^j$, three noisy labels $\{\bar{Y}^j, \bar{Y}^i, \bar{Y}^k\}$ will not suffice to identify $\boldsymbol{T}^j$.*

The inspiration from Theorem 3 is that, the increase of the number of noisy labels may make the identifiability decrease due to the entangled correlations (see Appendix G and I). To handle this problem, we prove Theorem 4 by assuming the transition relationship between noisy label for class $i$ and clean label for class $j$ is known.

**Theorem 4.** *If $P(\bar{Y}^i \mid Y^j)$ is known, two noisy labels $\{\bar{Y}^j, \bar{Y}^i\}$ are sufficient to identify $\boldsymbol{T}^j$.*

Theorem 4 theoretically guarantees that the identifiability of the class-dependent transition matrix can be achieved by utilizing the occurrence probabilities $P(\bar{Y}^i, \bar{Y}^j)$ and $P(\bar{Y}^i \mid Y^j)$. Note that $P(\bar{Y}^i, \bar{Y}^j)$ can represent noisy label correlations, and $P(\bar{Y}^i \mid Y^j) = P(\bar{Y}^i \mid Y^i)P(Y^i \mid Y^j)$, which can imply clean label correlations. At a high level, *the mismatch of label correlations* implied in the occurrence probabilities can achieve the identifiability.

The detailed proof of Theorem 1-4 is provided in Appendix E-H.

### 3.2   Our estimator

Our estimator is based on Theorem 4, which needs extra information to estimate $P(\bar{Y}^i \mid Y^j)$. Recently, the *memorization effect* [4] of deep networks has received much attention in learning with noisy labels, which shows that deep networks will first memorize the training data with clean labels and then those with noisy labels. Prior works utilize this characteristic to develop *sample selection* methods [16, 13, 22, 17, 25], where we select some examples with more likely clean labels for each class $j$ respectively in the early learning phase. The selected examples can serve as the useful extra information that implies clean label correlations, through which we can achieve estimation via counting. However, when implementing sample-selection-based methods, a major concern is whether the sampling bias will lead to large estimation errors.

Generally speaking, according to the *memorization effect*, the sampling bias is about selecting easy examples for class $j$, which usually means these examples have easy-to-discriminate features. We can reasonably assume that given $Y^j$, the distribution of the features about class $j$ is biased, while the distribution of the features about another class $i$ is unbiased, i.e.,

$$P_{D_s^j}(\bar{Y}^i|Y^j) = \int P_{D_s^j}(\bar{Y}^i|\boldsymbol{X}^i)P_{D_s^j}(\boldsymbol{X}^i|Y^j)d\boldsymbol{x} = \int P_{\bar{D}_s^j}(\bar{Y}^i|\boldsymbol{X}^i)P_{\bar{D}_s^j}(\boldsymbol{X}^i|Y^j)d\boldsymbol{x} = P_{\bar{D}_s^j}(\bar{Y}^i|Y^j), \tag{1}$$

where $D_s^j$ is the distribution of $(\boldsymbol{X}, \boldsymbol{Y_s})$, $\boldsymbol{Y_s} = \{\bar{Y}^1, \bar{Y}^2, ..., Y^j, ..., \bar{Y}^q\}$, $\bar{D}_s^j$ is the biased distribution of $(\boldsymbol{X}, \boldsymbol{Y_s})$, and $\boldsymbol{X}^i$ is the part of $\boldsymbol{X}$, which represents all information about class $i$ appearing or

not. When the assumption is satisfied, the sample selection will not lead to large estimation errors on $P(\bar{Y}^i \mid Y^j)$, and it can converge to zero exponentially fast by counting [6]. In real-world scenarios, due to the complex label correlations, this assumption will not strictly hold. While, it may be roughly met when class labels $i$ and $j$ do not share the major discriminative features. Intuitively speaking, the classifying of simplest examples for one label is not easily affected by the presence or absence of other significantly different labels. Also, since most label pairs from typical real-world multi-label datasets are significantly different (see Appendix D), the assumption can be roughly hold for those label pairs in typical cases. In Section 4.1, our empirical results justify this by showing a little gap between the estimation error of our estimator with a biased sample selection and an unbiased one.

Based on the above discussions, we can approximate $P(\bar{Y}^i, \bar{Y}^j)$ and $P(\bar{Y}^i \mid Y^j)$ with frequency counting, and utilize the mismatch of label correlations implied in these occurrence probabilities to estimate the transition matrix. In this work, we propose to estimate transition matrices $\{\boldsymbol{T}^j\}_{j=1}^q$ by following two stages.

In the first stage, we utilize sample selection to obtain useful information that implies clean label correlations. Specifically, we train a classifier with the standard multi-label classification loss on noisy training examples $\mathcal{D}_t$ for a few epochs, and then perform sample selection to get a selected clean set $\mathcal{D}_s^j$ for each class label $j$. Specially, we use a commonly used sample selection way [3, 22] in learning with noisy labels, which extracts the subset of examples with small losses by modeling the distribution of losses for class $j$ with a Gaussian mixture model (GMM).

In the second stage, we perform co-occurrence estimation by frequency counting, and then estimate the transition matrix by solving a simple bilinear decomposition problem. For class label $j$, we first choose another class label $i$ and estimate $P(\bar{Y}^i, \bar{Y}^j)$ and $P(\bar{Y}^i \mid Y^j)$ via counting, i.e.,

$$\hat{P}(\bar{Y}^i = v, \bar{Y}^j = k) = \frac{1}{n} \sum_{(\boldsymbol{x}, \bar{\boldsymbol{y}}) \in \mathcal{D}_t} \mathbb{I}[\bar{y}^j = v, \bar{y}^i = k] \quad \text{and} \tag{2}$$

$$\hat{P}(\bar{Y}^i = v \mid Y^j = k) = \frac{\sum_{(\boldsymbol{x}, \bar{\boldsymbol{y}}) \in \mathcal{D}_s^j} \mathbb{I}[\bar{y}^i = v, \bar{y}^j = k]}{\sum_{(\boldsymbol{x}, \bar{\boldsymbol{y}}) \in \mathcal{D}_s^j} \mathbb{I}[\bar{y}^j = k]}, \tag{3}$$

where $\mathbb{I}[\cdot]$ is the indicator function which takes 1 if the identity index is true and 0 otherwise.

Then, these co-occurrence probabilities, which imply the mismatch of label correlations, can lead to four equations involving $\boldsymbol{T}^j$:

$$P\left(\bar{Y}^j = 0, \bar{Y}^i = 0\right) = P(Y^j = 0)T_{00}^j P(\bar{Y}^i = 0 | Y^j = 0) + P(Y^j = 1)T_{10}^j P(\bar{Y}^i = 0 | Y^j = 1),$$

$$P\left(\bar{Y}^j = 0, \bar{Y}^i = 1\right) = P(Y^j = 0)T_{00}^j P(\bar{Y}^i = 1 | Y^j = 0) + P(Y^j = 1)T_{10}^j P(\bar{Y}^i = 1 | Y^j = 1),$$

$$P\left(\bar{Y}^j = 1, \bar{Y}^i = 0\right) = P(Y^j = 0)T_{01}^j P(\bar{Y}^i = 0 | Y^j = 0) + P(Y^j = 1)T_{11}^j P(\bar{Y}^i = 0 | Y^j = 1),$$

$$P\left(\bar{Y}^j = 1, \bar{Y}^i = 1\right) = P(Y^j = 0)T_{01}^j P(\bar{Y}^i = 1 | Y^j = 0) + P(Y^j = 1)T_{11}^j P(\bar{Y}^i = 1 | Y^j = 1).$$

For simplicity, we denote

$$\boldsymbol{E} = \begin{pmatrix} P(\bar{Y}^j = 0, \bar{Y}^i = 0) & P(\bar{Y}^j = 0, \bar{Y}^i = 1) \\ P(\bar{Y}^j = 1, \bar{Y}^i = 0) & P(\bar{Y}^j = 1, \bar{Y}^i = 1) \end{pmatrix} = \begin{pmatrix} e_{00} & e_{01} \\ e_{10} & e_{11} \end{pmatrix},$$

$$\boldsymbol{P} = \begin{pmatrix} P(Y^j = 0) & 0 \\ 0 & P(Y^j = 1) \end{pmatrix} = \begin{pmatrix} 1 - p & 0 \\ 0 & p \end{pmatrix},$$

$$\boldsymbol{T}^j = \begin{pmatrix} P(\bar{Y}^j = 0 \mid Y^j = 0) & P(\bar{Y}^j = 1 \mid Y^j = 0) \\ P(\bar{Y}^j = 0 \mid Y^j = 1) & P(\bar{Y}^j = 1 \mid Y^j = 1) \end{pmatrix} = \begin{pmatrix} 1 - \rho_- & \rho_- \\ \rho_+ & 1 - \rho_+ \end{pmatrix}, \quad \text{and}$$

$$\boldsymbol{M} = \begin{pmatrix} P(\bar{Y}^i = 0 \mid Y^j = 0) & P(\bar{Y}^i = 1 \mid Y^j = 0) \\ P(\bar{Y}^i = 0 \mid Y^j = 1) & P(\bar{Y}^i = 1 \mid Y^j = 1) \end{pmatrix} = \begin{pmatrix} 1 - \rho'_- & \rho'_- \\ \rho'_+ & 1 - \rho'_+ \end{pmatrix}.$$

Then the system of equations can be expressed as $\boldsymbol{E} = (\boldsymbol{T}^j)^\top \boldsymbol{P} \boldsymbol{M}$, i.e.,

$$\begin{pmatrix} e_{00} & e_{01} \\ e_{10} & e_{11} \end{pmatrix} = \begin{pmatrix} 1 - \rho_- & \rho_- \\ \rho_+ & 1 - \rho_+ \end{pmatrix}^\top \begin{pmatrix} 1 - p & 0 \\ 0 & p \end{pmatrix} \begin{pmatrix} 1 - \rho'_- & \rho'_- \\ \rho'_+ & 1 - \rho'_+ \end{pmatrix}.$$

Denote the estimation of $\boldsymbol{E}, \boldsymbol{P}, \boldsymbol{T}^j$, and $\boldsymbol{M}$ as

$$\hat{\boldsymbol{E}} = \begin{pmatrix} \hat{e}_{00} & \hat{e}_{01} \\ \hat{e}_{10} & \hat{e}_{11} \end{pmatrix}, \hat{\boldsymbol{P}} = \begin{pmatrix} 1 - \hat{p} & 0 \\ 0 & \hat{p} \end{pmatrix}, \hat{\boldsymbol{T}}^j = \begin{pmatrix} 1 - \hat{\rho}_- & \hat{\rho}_- \\ \hat{\rho}_+ & 1 - \hat{\rho}_+ \end{pmatrix}, \hat{\boldsymbol{M}} = \begin{pmatrix} 1 - \hat{\rho}'_- & \hat{\rho}'_- \\ \hat{\rho}'_+ & 1 - \hat{\rho}'_+. \end{pmatrix}$$

As $\hat{E}$ and $\hat{M}$ can be derived from Eq. (2) and Eq. (3), the problem is hence equivalent to a bilinear decomposition problem:

$$\hat{E}(\hat{M})^{-1} = (\hat{T}^j)^\top \hat{P}. \tag{4}$$

By solving the above matrix equation, we can get

$$\hat{p} = \frac{(1 - \hat{\rho}'_-) - (\hat{e}_{00} + \hat{e}_{10})}{1 - \hat{\rho}'_- - \hat{\rho}'_+}, \tag{5}$$

and the estimation of the transition matrix

$$\hat{T}^j = [\hat{E}(\hat{M})^{-1}(\hat{P})^{-1}]^\top. \tag{6}$$

**Implementation of our estimator.** The pseudo code of our estimator is described in Algorithm 1. A little difference from the above is that in order to make better use of correlations among labels, we perform $R$ times co-occurrence estimation and bilinear decomposition for different classes $i$ in the second stage to get $R$ estimations, $\hat{T}^j_r, r = 1, 2, ..., R$. Finally, we estimate the transition matrix $T^j$ by Eq. (7):

$$\hat{T}^j = \arg\min_{\hat{T}^j_r} \sum_{i=1}^R \|\hat{T}^j_r - \hat{T}^j_i\|_1, \tag{7}$$

where $\| \cdot \|_1$ denotes $\ell_1$ norm.

---

**Algorithm 1** Estimating Label-Noise Transition Matrices using Label Correlations

---

**Require:** Noisy training examples $\mathcal{D}_t$, the number of classes $q$, the early warmup training epoch $E_{warm}$, the threshold of sample selection $\tau$, and repeated estimation times $R$.
    **Stage1: Standard Training and Sample Selection**
1: Standard training with the standard multi-label loss for $E_{warm}$ epochs.
2: **for** $j = 1, 2, ..., q$ **do**
3:     Model losses with a trained classifier on $\mathcal{D}_t$ by a GMM.
4:     Get the selected set $\mathcal{D}^j_s$ for class label $j$ with the threshold $\tau$.
5: **end for**
    **Stage2: Co-occurrence Estimation and Bilinear Decomposition**
6: **for** $j = 1, 2, ..., q$ **do**
7:     **for** $r = 1, 2, ..., R$ **do**
8:         Choose another class label $i$.
9:         Estimate $P(\bar{Y}^i, \bar{Y}^j)$ by $\hat{P}(\bar{Y}^i, \bar{Y}^j)$ with Eq. (2) on $\mathcal{D}_t$, and $P(\bar{Y}^i \mid Y^j)$ by $\hat{P}(\bar{Y}^i \mid Y^j)$ with Eq. (3) on $\mathcal{D}^j_s$.
10:         Solve a bilinear decomposition problem (Eq. (4)) to get a estimation $\hat{T}^j_r$ by Eq. (6).
11:     **end for**
12:     Estimate $T^j$ by $\hat{T}^j$ which has the minimum error with Eq. (7) from $R$ estimations.
13: **end for**
**Ensure:** The estimated transition matrices $\{\hat{T}^j\}^q_{j=1}$.

---

## 4 Experiments

**Dataset** We verify the effectiveness of the proposed method on three synthetic noisy multi-label datasets, i.e., Pascal-VOC2007 [11], Pascal-VOC2012 [12], and MS-COCO [27]. Pascal-VOC2007 [11] and Pascal-VOC2012 [12] datasets are two popular datasets for object recognition. They both contain images from the same 20 object classes, with an average of $n_a = 1.5$ labels per image. Pascal-VOC2007 contains a training set of 5,011 images and a test set of 4,952 images. Pascal-VOC2012 consists of 11,540 images as the training set and 10,991 images as the test set [5]. As the labels of the test set in Pascal-VOC2012 are not publicly available, we use the test set in Pascal-VOC2007 for Pascal-VOC2012 evaluation. MS-COCO [27] is a widely used multi-label dataset. It contains 82,081 images as the training set and 40,137 images as the test set and covers 80 object classes with an average of $n_a = 2.9$ labels per image. For these datasets, we corrupted the training sets manually according to true transition matrices $\{T^j\}^q_{j=1}$. For convenience, we use the

same true transition matrices for all classes, i.e., $\boldsymbol{T}^j = \boldsymbol{T} = \begin{pmatrix} 1 - \rho_- & \rho_- \\ \rho_+ & 1 - \rho_+ \end{pmatrix}$, but do not divulge this information for algorithms. We generate four different types of synthetic datasets by using different transition matrices: 1) $\rho_- = 0, \rho_+ = \rho$, which annotates some positive examples as negative examples, also known as multi-label learning with missing labels [45, 44]; 2) $\rho_- = \rho, \rho_+ = 0$, which annotates some negative examples as positive examples, also known as partial multi-label learning [53, 55]; 3) $\rho_- = \rho, \rho_+ = \rho$, where positive examples and negative examples are mislabeled with the same probability $\rho$; 4) $\rho_- = \frac{n_a}{q - n_a}\rho, \rho_+ = \rho$, where positive examples and negative examples are mislabeled with the same number. In the experiments, we test the algorithms on various $\rho$. For all datasets, we leave out 10% of the noisy training examples as a noisy validation set. We use mAP on noisy validation set as the criterion for model selection.

**Implementation details** For a fair comparison, we implement all methods with default parameters by PyTorch on NVIDIA RTX 3090. We use a ResNet-50 network [14] pre-trained on ImageNet [37] for all datasets, and the optimizer is Adam optimizer [18] with $\beta = (0.9, 0.999)$. The batch size is 128, the learning rate is 5e-5. The number of training epochs is 20 for Pascal-VOC2007/VOC2012, and 30 for MS-COCO. For the transition matrix estimation method, $E_{warm}$ is the same as the normal training epoch. For our estimator, we perform sample selection based on the average losses of 5 epochs before a certain warmup epoch (10th epoch for Pascal-VOC2007/VOC2012, 15th epoch for MS-COCO), $R = q - 1$ and $\tau = 0.5$ in all experiments. All experiments are run at least three times with different random seeds, and we report the average and standard deviation values of results. The best results are in **bold**, and the second-best results are in blue.

### 4.1 Comparison for estimating transition matrices

**Baselines** For evaluating the effectiveness of estimating the transition matrix under multi-label cases, we compare the proposed method with the following methods: (1) T-estimator max [28, 33], which estimates the transition matrix via the noisy class posterior probabilities for anchor points that have the largest estimated noisy class posteriors. (2) T-estimator 97% [28, 33], which selects the points with 97% largest estimated noisy class posteriors to be anchor points. (3) Dual T-estimator max [57], which introduces an intermediate class to avoid directly estimating the noisy class posterior, and selects the points with the largest estimated intermediate class posteriors to be the anchor points. (4) Dual T-estimator 97% [57], which selects the points with the 97% largest estimated intermediate class posteriors to be the anchor points.

**Metrics** We use the sum of estimation error for the transition matrices as the estimation evaluation metric, i.e., $\sum_{j=1}^{q} \|\boldsymbol{T}^j - \hat{\boldsymbol{T}}^j\|_1 / \|\boldsymbol{T}^j\|_1$.

**Results** In Tab. 1, 2 and 3, we can see that for all cases on three datasets, the proposed estimation method leads to the smallest or second-smallest estimator errors across various noise rates. Note that since the fitting of noisy or intermediate class posterior is hard to be accurate in noisy multi-label learning, T-estimator and Dual T-estimator need to carefully tune a hyperparameter for better estimation under different noise rates, and it's very sensitive in some cases, e.g., MS-COCO datasets with noise rates $(0.1, 0.1)$. In contrast, our method uses the same hyperparameters on one dataset to get good results for all cases, which reflects its robustness to various noise rates. Besides, to study the ablation of sampling bias, we also run our method with an unbiased sample selection, named "our estimator gold". We can see that sample bias is the main factor that contributes to the error for our estimator, but the little error gap between our estimator and our estimator gold shows it will not lead to large estimation errors.

Table 1: Comparison for estimating transition matrices on Pascal-VOC2007 dataset.

| Noise rates $(\rho_-, \rho_+)$ | (0,0.2) | (0,0.6) | (0.2,0) | (0.6,0) | (0.1,0.1) | (0.2,0.2) | (0.017,0.2) | (0.034,0.4) |
|---|---|---|---|---|---|---|---|---|
| T-estimator max | 3.89±0.03 | 10.52±0.58 | 3.01±0.12 | 4.47±0.22 | 3.18±0.22 | 5.28±0.20 | 3.99±0.10 | 6.28±0.44 |
| T-estimator 97% | 4.95±0.17 | 4.42±0.18 | 1.77±0.03 | 2.13±0.12 | 6.99±0.10 | 6.94±0.17 | 5.38±0.14 | 5.17±0.09 |
| Dual T-estimator max | 1.94±0.13 | 7.29±0.16 | 1.03±0.04 | 2.68±0.13 | **2.13±0.23** | 4.02±0.18 | **1.71±0.08** | 2.67 ±0.27 |
| Dual T-estimator 97% | 12.59±0.06 | 7.43±0.06 | 1.09±0.03 | 2.41±0.33 | 14.39±0.10 | 11.78±0.06 | 13.71±0.16 | 11.15±0.09 |
| Our estimator | **1.51±0.12** | **2.30±0.13** | **0.37±0.08** | **1.34±0.33** | 3.06±0.38 | **3.21±0.32** | 2.03±0.19 | **1.84±0.32** |
| Our estimator gold | 0.44±0.05 | 0.51±0.09 | 0.38±0.08 | 0.37±0.11 | 0.83±0.05 | 2.15±0.30 | 0.65±0.10 | 1.40±0.20 |

Table 2: Comparison for estimating transition matrices on Pascal-VOC2012 dataset.

| Noise rates $(\rho_-, \rho_+)$ | (0,0.2) | (0,0.6) | (0.2,0) | (0.6,0) | (0.1,0.1) | (0.2,0.2) | (0.017,0.2) | (0.034,0.4) |
|---|---|---|---|---|---|---|---|---|
| T-estimator max | 3.90±0.01 | 10.28±0.33 | 2.87±0.09 | 4.55±0.08 | 3.29±0.07 | 5.25±0.15 | 4.05±0.04 | 6.82±0.20 |
| T-estimator 97% | 5.42±0.09 | 3.98±0.09 | 1.53±0.06 | 1.91±0.07 | 6.43±0.16 | 6.20±0.17 | 5.76±0.27 | 5.16±0.14 |
| Dual T-estimator max | 1.02±0.20 | 5.13±0.26 | 1.07±0.07 | 2.06±0.12 | 1.94±0.05 | 2.59±0.16 | 1.17±0.13 | 1.93±0.08 |
| Dual T-estimator 97% | 12.94±0.06 | 7.49±0.03 | 1.14±0.04 | 2.94±0.18 | 14.23±0.08 | 11.56±0.05 | 13.97±0.09 | 11.10±0.08 |
| Our estimator | **0.83±0.10** | **1.94±0.15** | **0.26±0.03** | **0.91±0.12** | **1.74±0.22** | **1.79±0.17** | **0.94±0.07** | **1.07±0.14** |
| Our estimator gold | 0.33±0.05 | 0.34±0.05 | 0.25±0.05 | 0.45±0.05 | 0.51±0.05 | 1.67±0.29 | 0.42±0.06 | 0.91±0.16 |

Table 3: Comparison for estimating transition matrices on MS-COCO dataset.

| Noise rates $(\rho_-, \rho_+)$ | (0,0.2) | (0,0.6) | (0.2,0) | (0.6,0) | (0.1,0.1) | (0.2,0.2) | (0.008,0.2) | (0.015,0.4) |
|---|---|---|---|---|---|---|---|---|
| T-estimator max | 16.14±0.33 | 39.09±0.47 | 10.39±0.21 | 11.49±0.60 | 13.95±0.41 | 20.50±0.04 | 16.70±0.06 | 28.16±0.45 |
| T-estimator 97% | 50.49±0.01 | 25.70±0.08 | 4.04±0.08 | 3.70±0.02 | 51.17±0.16 | 39.45±0.11 | 49.96±0.18 | 37.54±0.10 |
| Dual T-estimator max | **5.04±0.04** | **11.22±0.70** | 4.65±0.07 | 9.55±0.84 | 13.02±0.45 | 15.79±0.38 | **7.04±0.31** | **6.34±0.11** |
| Dual T-estimator 97% | 61.49±0.02 | 30.97±0.03 | 1.53±0.00 | 7.86±0.12 | 64.20±0.02 | 48.67±0.01 | 63.12±0.02 | 46.91±0.01 |
| Our estimator | 7.42±0.38 | 11.23±0.11 | **0.50±0.03** | **0.83±0.06** | **8.88±0.10** | **10.27±0.19** | 7.51±0.43 | 8.77±0.20 |
| Our estimator gold | 0.82±0.03 | 0.80±0.04 | 0.40±0.04 | 0.66±0.04 | 1.94±0.04 | 8.14±0.06 | 0.95±0.05 | 2.02±0.08 |

## 4.2 Comparison for classification performance

**Baselines** We exploit 10 baselines: (1) Standard, which trains with a standard multi-label classification loss. (2) GCE [60], which proposes a Generalized Cross-Entropy loss for robustness. (3) CDR [49], which performs different update rules for two types of parameters to achieve robust learning. (4) AGCN [58], which adopts a dynamic GCN to model the relation of content-aware class representations. (5) CSRA [61], which generates class-specific features for every category by proposing a spatial attention score. (6) WSIC [15] that proposes to use a small set with clean labels to learn a residual net for regularization in noisy multi-label learning, and we only provide noisy datasets to it for a fair comparison. (7) Reweight-T max, which learns with a reweighting algorithm using transition matrices estimated by T-estimator max [33]. (8) Reweight-T 97%, which learns with a reweighting algorithm using transition matrices estimated by T-estimator 97% [33]. (9) Reweight-DualT max, which learns with a reweighting algorithm using transition matrices estimated by Dual T-estimator max [57]. (10) Reweight-DualT 97%, which learns with a reweighting algorithm using transition matrices estimated by Dual T-estimator 97% [57]. Note that Standard, AGCN, and CSRA are designed for clean multi-label data, and GCE and CDR are designed for noisy multi-class learning.

**Metrics** Following conventional setting [11, 7, 36, 35], we compute the mean average precision (mAP), overall F1-measure (OF1) and per-class F1-measure (CF1) as classification evaluation metrics. For each image, we assign a positive label if its prediction probability is greater than 0.5.

**Results** As shown in Tab. 4, 5 and 6, first, we can find those statistically consistent methods achieve the best or second-best results on all three metrics in the vast majority of cases, while other methods can only achieve good results in some cases. For example, on the Pascal-VOC2012 dataset with noise rates $(0.0, 0.6)$, CSRA achieves the best result in the mAP metric with the help of its well-designed network, but its performance is far lower than those consistent methods on the OF1 and CF1 metrics, which shows the learned model can not approximate well the true class posterior $P(\boldsymbol{Y}|\boldsymbol{X})$. Note that since network structure and loss correction are compatible, the risk-consistent methods can also help AGCN and CSRA perform better (shown in Appendix M). Second, theoretically, the more accurate the transition matrix is estimated, the more likely the consistent method is to achieve better results by increasing the size of noisy examples, and our experimental results verify this. Among those consistent methods, the reweighting algorithm with our estimator (named "Reweight-Ours") obtains the best or second-best results on the three metrics, which is due to the smaller error of our estimation. Especially on the challenging and large-scale MS-COCO dataset, Reweight-Ours outperforms almost all state-of-the-art methods on the CF1 metric and significantly surpasses other baselines by a large margin in some cases. For example, on the MS-COCO dataset with noise rates $(0.6, 0)$, Reweight-Ours achieves the best CF1 result (58.63±1.30), while the suboptimal result is 56.70±2.11, and the result of Standard is only 7.07±0.02. In addition, the ablation studies about loss correction ways and base learning algorithms are provided in Appendix L and M, which shows that our estimator can achieve much better performance with the advanced frameworks.

Besides, although our method is based on the assumption of class-dependent label noise, the experiments with two types of instance-dependent label noise are provided in Appendix O.

Table 4: Comparison for classification performance on Pascal-VOC2007 dataset.

| | Noise rates ($\rho_-$, $\rho_+$) | (0,0.2) | (0,0.6) | (0.2,0) | (0.6,0) | (0.1,0.1) | (0.2,0.2) | (0.017,0.2) | (0.034,0.4) |
|---|---|---|---|---|---|---|---|---|---|
| mAP | Standard | 84.25±1.07 | 77.16±0.94 | 82.70±0.54 | 68.65±1.57 | 83.07±0.45 | 78.87±0.52 | 83.92±0.59 | 80.97±0.42 |
| | GCE | 83.85±1.09 | 73.32±2.22 | 83.03±0.51 | 67.47±1.74 | 83.68±0.66 | 79.39±0.95 | 84.40±0.34 | 80.68±0.52 |
| | CDR | 84.60±0.43 | 77.45±1.23 | 82.76±0.53 | 68.86±2.05 | 83.22±0.57 | 79.02±0.62 | 84.37±0.25 | 81.14±0.28 |
| | AGCN | 83.24±0.67 | 75.50±0.56 | 81.09±0.51 | 66.47±1.29 | 81.09±0.48 | 73.79±0.76 | 82.21±0.42 | 76.55±1.11 |
| | CSRA | 85.11±0.51 | 79.47±1.22 | 82.93±0.65 | 67.36±2.25 | 83.69±0.69 | 78.10±0.53 | 84.94±0.36 | 81.51±0.14 |
| | WSIC | 84.14±0.26 | 76.17±1.31 | 82.30±0.64 | 66.82±3.87 | 83.41±0.31 | 77.93±1.00 | 84.17±0.48 | 80.74±0.44 |
| | Reweight-T max | 84.20±0.46 | 76.97±1.20 | 83.04±0.39 | 71.36±2.47 | 83.48±0.15 | 79.10±0.52 | 84.06±0.24 | 81.01±0.99 |
| | Reweight-T 97% | 84.00±0.68 | 78.97±0.69 | 83.07±0.29 | 73.96±1.69 | 82.71±0.30 | 78.80±0.28 | 84.37±0.22 | 81.42±0.25 |
| | Reweight-DualT max | 84.46±0.20 | 77.65±1.06 | 83.75±0.44 | 73.75±1.61 | 83.94±0.31 | 79.48±1.24 | 84.60±0.30 | 81.77±0.26 |
| | Reweight-DualT 97% | 82.36±0.45 | 77.72±0.73 | 84.56±0.40 | 75.76±2.11 | 79.69±1.40 | 75.26±1.70 | 81.84±0.81 | 77.40±1.86 |
| | Reweight-Ours | 84.43±0.46 | 78.72±0.41 | 84.08±0.24 | 74.46±0.56 | 84.03±0.29 | 80.44±0.52 | 84.09±0.62 | 80.97±1.03 |
| OF1 | Standard | 75.24±1.40 | 32.02±5.49 | 78.85±0.43 | 15.08±0.25 | 79.24±0.43 | 75.85±0.84 | 75.98±1.04 | 59.67±1.65 |
| | GCE | 76.17±1.57 | 36.13±4.07 | 79.28±0.44 | 14.85±0.22 | 79.73±0.70 | 76.27±0.55 | 76.80±0.68 | 60.26±2.43 |
| | CDR | 76.05±0.68 | 34.11±3.43 | 79.04±0.46 | 14.99±0.19 | 79.34±0.60 | 76.00±0.47 | 76.56±0.52 | 59.31±1.04 |
| | AGCN | 74.92±1.02 | 30.97±3.78 | 75.45±2.06 | 16.85±0.56 | 78.69±0.31 | 72.64±0.51 | 75.16±0.58 | 56.56±1.64 |
| | CSRA | 76.94±1.03 | 33.65±2.73 | 77.71±1.23 | 15.94±0.32 | 80.36±0.53 | 76.92±0.34 | 77.91±0.63 | 62.19±1.97 |
| | WSIC | 75.01±1.18 | 16.48±6.78 | 79.02±0.59 | 14.88±0.21 | 78.55±1.05 | 72.88±3.44 | 72.30±2.82 | 53.26±9.44 |
| | Reweight-T max | 76.97±0.45 | 41.54±2.64 | 79.65±0.44 | 47.68±5.65 | 80.00±0.27 | 73.58±1.67 | 76.94±0.37 | 66.77±0.93 |
| | Reweight-T 97% | 77.71±0.65 | 68.28±2.03 | 80.16±0.24 | 70.67±0.70 | 75.28±0.97 | 65.03±2.20 | 78.45±0.63 | 74.11±0.67 |
| | Reweight-DualT max | 78.38±0.41 | 68.81±1.41 | 80.02±1.12 | 65.41±1.84 | 79.87±0.27 | 65.36±7.31 | 78.55±0.36 | 47.04±4.15 |
| | Reweight-DualT 97% | 68.17±3.53 | 61.81±3.29 | 80.74±0.50 | 72.53±2.65 | 52.99±5.06 | 36.75±6.71 | 67.55±0.64 | 57.41±0.70 |
| | Reweight-Ours | 78.62±0.58 | 65.68±1.67 | 80.85±0.25 | 67.43±4.65 | 79.64±0.29 | 75.52±0.86 | 79.25±0.52 | 74.35±1.65 |
| CF1 | Standard | 72.53±1.11 | 30.64±3.90 | 76.83±0.65 | 14.97±0.24 | 75.86±1.23 | 70.68±1.76 | 73.11±0.54 | 52.07±2.34 |
| | GCE | 73.10±1.27 | 33.07±4.65 | 77.25±0.66 | 14.77±0.20 | 76.73±1.57 | 71.24±1.42 | 73.37±0.98 | 56.89±2.84 |
| | CDR | 73.08±0.47 | 33.06±1.84 | 76.95±0.79 | 14.88±0.17 | 76.09±1.42 | 70.78±1.09 | 73.33±0.85 | 54.16±3.19 |
| | AGCN | 73.45±1.04 | 33.41±1.65 | 72.65±1.97 | 16.67±0.55 | 76.20±0.51 | 69.09±0.49 | 72.81±1.02 | 55.09±3.28 |
| | CSRA | 74.10±0.56 | 33.44±3.65 | 75.28±1.32 | 15.71±0.23 | 77.52±0.94 | 73.44±0.62 | 74.98±0.48 | 58.60±2.24 |
| | WSIC | 70.13±2.04 | 13.64±6.97 | 75.32±2.00 | 14.77±0.16 | 74.17±1.87 | 64.95±6.47 | 65.41±4.40 | 43.43±11.31 |
| | Reweight-T max | 74.05±0.51 | 39.37±1.36 | 77.28±0.47 | 50.35±5.52 | 77.20±0.39 | 73.32±0.47 | 74.08±0.29 | 62.99±1.80 |
| | Reweight-T 97% | 76.81±0.74 | 71.24±1.33 | 77.22±0.56 | 67.65±1.37 | 74.99±0.37 | 68.07±0.74 | 77.62±0.42 | 74.56±0.26 |
| | Reweight-DualT max | 75.27±0.56 | 45.31±1.86 | 76.56±1.80 | 63.99±0.42 | 77.27±0.40 | 69.48±4.48 | 75.66±0.35 | 68.32±1.22 |
| | Reweight-DualT 97% | 71.93±1.64 | 63.85±3.69 | 77.95±0.79 | 68.44±2.93 | 62.84±2.51 | 50.49±2.43 | 70.99±0.98 | 60.58±1.67 |
| | Reweight-Ours | 76.86±0.48 | 61.29±1.94 | 77.89±0.42 | 66.79±2.50 | 78.04±0.40 | 74.08±0.79 | 77.28±0.48 | 72.18±0.74 |

Table 5: Comparison for classification performance on Pascal-VOC2012 dataset.

| | Noise rates ($\rho_-$, $\rho_+$) | (0,0.2) | (0,0.6) | (0.2,0) | (0.6,0) | (0.1,0.1) | (0.2,0.2) | (0.017,0.2) | (0.034,0.4) |
|---|---|---|---|---|---|---|---|---|---|
| mAP | Standard | 85.97±0.09 | 80.02±0.62 | 85.70±0.19 | 76.13±0.86 | 85.91±0.10 | 82.54±0.51 | 86.03±0.24 | 82.91±0.74 |
| | GCE | 86.02±0.21 | 78.71±0.72 | 85.96±0.19 | 75.02±0.50 | 86.29±0.15 | 83.18±0.33 | 85.84±0.37 | 82.96±0.83 |
| | CDR | 86.09±0.14 | 80.52±1.61 | 85.61±0.18 | 76.53±1.26 | 86.01±0.19 | 82.79±0.42 | 85.92±0.38 | 83.48±0.62 |
| | AGCN | 85.16±0.12 | 79.67±0.43 | 84.91±0.38 | 77.54±0.29 | 84.69±0.30 | 80.15±0.49 | 84.75±0.12 | 81.85±0.35 |
| | CSRA | 86.88±0.23 | 81.75±0.97 | 85.39±0.27 | 75.08±0.84 | 86.14±0.14 | 80.98±1.22 | 86.51±0.15 | 83.86±0.49 |
| | WSIC | 86.39±0.38 | 80.75±0.49 | 85.53±0.28 | 77.00±1.03 | 85.67±0.17 | 82.01±0.87 | 86.07±0.22 | 83.19±0.19 |
| | Reweight-T max | 85.40±0.43 | 79.06±1.69 | 85.80±0.32 | 78.59±0.69 | 85.98±0.32 | 82.88±0.41 | 85.51±0.51 | 82.38±1.67 |
| | Reweight-T 97% | 85.97±0.27 | 81.04±0.79 | 85.81±0.21 | 80.12±0.75 | 85.66±0.55 | 82.81±0.52 | 85.99±0.51 | 83.28±1.07 |
| | Reweight-DualT max | 85.93±0.41 | 78.69±2.62 | 86.47±0.26 | 80.00±0.66 | 86.23±0.34 | 84.21±0.18 | 86.15±0.44 | 83.42±1.26 |
| | Reweight-DualT 97% | 83.93±0.52 | 79.97±1.42 | 86.37±0.25 | 81.84±0.87 | 82.39±0.85 | 78.61±1.01 | 83.33±1.40 | 80.95±1.15 |
| | Reweight-Ours | 86.01±0.54 | 80.33±1.85 | 86.12±0.10 | 80.54±1.30 | 86.23±0.28 | 83.42±0.51 | 85.92±0.42 | 83.56±1.31 |
| OF1 | Standard | 77.91±0.20 | 27.84±4.17 | 80.47±0.34 | 14.93±0.28 | 81.27±0.22 | 78.51±0.13 | 77.83±0.36 | 61.67±2.10 |
| | GCE | 78.25±0.28 | 24.67±3.13 | 80.89±0.19 | 14.73±0.28 | 81.47±0.34 | 78.78±0.11 | 78.32±0.37 | 63.26±1.40 |
| | CDR | 78.02±0.23 | 29.45±6.39 | 80.65±0.36 | 14.91±0.26 | 81.34±0.35 | 78.58±0.14 | 77.82±0.34 | 61.94±1.91 |
| | AGCN | 76.11±0.76 | 31.06±4.73 | 80.49±0.51 | 15.42±0.35 | 80.28±0.44 | 76.02±1.33 | 75.83±1.18 | 61.59±4.20 |
| | CSRA | 78.61±0.56 | 32.36±6.27 | 79.88±0.57 | 15.71±0.50 | 81.82±0.35 | 78.12±1.11 | 79.05±0.59 | 61.77±3.46 |
| | WSIC | 78.56±0.78 | 27.90±6.72 | 80.56±0.32 | 15.09±0.24 | 80.98±0.49 | 77.36±1.51 | 78.52±0.65 | 57.56±4.11 |
| | Reweight-T max | 77.76±0.72 | 43.96±6.52 | 81.24±0.31 | 48.88±2.45 | 81.53±0.49 | 78.63±0.17 | 78.66±0.22 | 68.04±2.70 |
| | Reweight-T 97% | 79.31±0.20 | 71.79±2.35 | 81.56±0.29 | 75.10±1.81 | 77.81±1.09 | 71.28±1.99 | 79.20±0.35 | 73.90±1.81 |
| | Reweight-DualT max | 80.44±0.69 | 63.68±5.75 | 82.01±0.28 | 74.17±2.34 | 81.04±0.36 | 78.71±0.83 | 80.47±0.35 | 75.56±1.72 |
| | Reweight-DualT 97% | 60.81±1.36 | 58.42±2.41 | 82.20±0.20 | 75.96±1.47 | 56.42±1.24 | 30.28±1.82 | 64.36±1.42 | 58.49±1.73 |
| | Reweight-Ours | 80.54±0.61 | 69.08±4.58 | 81.77±0.43 | 75.38±3.05 | 81.11±0.56 | 77.78±0.37 | 80.75±0.43 | 77.31±1.49 |
| CF1 | Standard | 75.15±0.62 | 29.76±3.77 | 79.06±0.35 | 14.87±0.29 | 79.05±0.26 | 75.61±0.52 | 76.07±0.46 | 60.32±1.90 |
| | GCE | 75.25±0.84 | 26.25±4.40 | 79.68±0.32 | 14.69±0.27 | 79.13±0.32 | 75.84±0.51 | 76.05±0.84 | 61.99±0.71 |
| | CDR | 75.32±0.66 | 31.49±4.03 | 79.20±0.20 | 14.85±0.25 | 79.14±0.45 | 75.56±0.49 | 76.11±0.44 | 60.02±1.91 |
| | AGCN | 74.16±1.06 | 33.33±4.55 | 78.81±0.12 | 15.42±0.76 | 78.09±0.99 | 73.28±1.80 | 73.91±1.39 | 58.57±4.67 |
| | CSRA | 75.91±0.98 | 32.83±4.88 | 78.86±0.25 | 15.48±0.49 | 79.87±0.26 | 75.53±1.12 | 76.40±0.73 | 59.53±3.12 |
| | WSIC | 76.46±1.39 | 30.19±5.09 | 79.49±0.57 | 14.98±0.23 | 78.50±0.91 | 74.10±2.39 | 76.05±0.90 | 55.11±4.30 |
| | Reweight-T max | 75.54±0.59 | 39.66±5.13 | 79.79±0.27 | 49.60±3.31 | 79.60±0.48 | 76.45±0.29 | 76.82±0.42 | 65.53±2.66 |
| | Reweight-T 97% | 78.94±0.10 | 73.66±1.10 | 79.73±0.29 | 73.38±1.41 | 77.54±0.88 | 72.39±1.57 | 78.78±0.35 | 75.54±1.13 |
| | Reweight-DualT max | 78.37±0.54 | 58.96±3.99 | 80.06±0.22 | 72.61±1.76 | 79.29±0.35 | 76.37±1.44 | 78.69±0.25 | 65.68±1.37 |
| | Reweight-DualT 97% | 67.96±1.13 | 61.89±2.49 | 80.11±0.33 | 71.87±3.50 | 65.09±1.07 | 52.28±1.03 | 69.26±0.82 | 63.10±1.57 |
| | Reweight-Ours | 78.81±0.53 | 66.76±3.38 | 79.82±0.43 | 72.84±2.16 | 79.90±0.39 | 75.90±0.81 | 79.37±0.35 | 75.16±1.42 |

Table 6: Comparison for classification performance on MS-COCO dataset.

| | Noise rates $(\rho_-, \rho_+)$ | (0,0.2) | (0,0.6) | (0.2,0) | (0.6,0) | (0.1,0.1) | (0.2,0.2) | (0.008,0.2) | (0.015,0.4) |
|---|---|---|---|---|---|---|---|---|---|
| mAP | Standard | 69.92±0.06 | 63.81±0.16 | 66.77±0.52 | 55.45±0.48 | 67.77±0.28 | 62.50±0.23 | 69.76±0.09 | 66.82±0.05 |
| | GCE | 69.90±0.05 | 62.58±0.17 | 67.32±0.11 | 54.01±0.70 | 68.62±0.16 | 63.21±0.35 | 69.99±0.11 | 66.72±0.19 |
| | CDR | 70.06±0.05 | 63.85±0.28 | 67.32±0.08 | 55.20±1.62 | 68.01±0.08 | 62.65±0.21 | 69.87±0.09 | 66.85±0.19 |
| | AGCN | **71.48±0.14** | **65.75±0.32** | 69.44±0.10 | 55.71±0.61 | **69.42±0.23** | 63.96±0.11 | **70.90±0.13** | 67.86±0.26 |
| | CSRA | 71.18±0.10 | 65.28±0.11 | 67.93±0.18 | 51.49±0.73 | 68.83±0.12 | 61.80±0.98 | 70.76±0.16 | **68.02±0.15** |
| | WSIC | 68.92±0.09 | 63.09±0.28 | 66.22±0.06 | 53.61±0.36 | 67.41±0.15 | 62.33±0.18 | 68.95±0.15 | 66.29±0.21 |
| | Reweight-T max | 69.99±0.18 | 63.94±0.11 | 67.40±0.13 | 58.27±0.25 | 67.85±0.05 | 63.28±0.12 | 69.76±0.07 | 66.24±0.51 |
| | Reweight-T 97% | 67.98±0.57 | 62.52±0.46 | 68.00±0.17 | 59.44±0.81 | 65.69±0.48 | 60.03±0.11 | 68.13±0.04 | 64.40±0.18 |
| | Reweight-DualT max | 67.57±0.21 | 60.39±0.53 | 68.57±0.25 | 58.42±0.82 | 68.01±0.51 | 62.17±0.32 | 68.76±0.08 | 65.75±0.15 |
| | Reweight-DualT 97% | 64.97±0.20 | 58.85±0.43 | **69.68±0.25** | 49.17±3.02 | 56.36±0.62 | 49.41±0.38 | 63.27±0.36 | 58.21±0.86 |
| | Reweight-Ours | 70.57±0.11 | 63.28±0.92 | 69.38±0.36 | **61.88±0.66** | 68.70±0.15 | **64.46±0.10** | 70.06±0.06 | 67.03±0.08 |
| OF1 | Standard | 66.48±0.50 | 19.18±0.97 | 69.58±0.38 | 7.05±0.01 | 68.64±0.18 | 64.84±0.54 | 66.07±0.15 | 51.70±0.43 |
| | GCE | 66.67±0.38 | 19.61±1.61 | 69.82±0.25 | 7.03±0.02 | 69.44±0.18 | 64.98±0.79 | 66.58±0.32 | 52.04±0.51 |
| | CDR | 66.46±0.54 | 19.60±2.86 | 69.72±0.31 | 7.06±0.04 | 68.75±0.09 | 64.68±0.54 | 66.03±0.39 | 52.49±1.18 |
| | AGCN | 67.14±0.52 | 16.02±0.76 | **70.63±0.12** | 7.04±0.02 | 69.66±0.25 | **66.07±0.55** | 66.61±0.48 | 52.38±1.05 |
| | CSRA | 67.98±0.40 | 24.08±2.60 | 70.14±0.17 | 7.06±0.02 | **69.70±0.31** | 64.89±1.22 | 67.37±0.28 | 51.81±0.53 |
| | WSIC | 66.67±0.15 | 23.02±4.70 | 69.02±0.06 | 7.02±0.00 | 67.78±0.58 | 62.31±0.69 | 66.38±0.25 | 52.07±1.94 |
| | Reweight-T max | 67.13±0.41 | 39.47±1.47 | 69.84±0.11 | 59.26±1.76 | 64.03±2.00 | 57.68±4.00 | 66.45±0.30 | 53.63±1.28 |
| | Reweight-T 97% | 57.66±0.56 | 54.06±1.19 | 69.72±0.39 | 64.79±0.85 | 43.81±0.54 | 33.65±2.93 | 55.44±0.56 | 51.78±1.37 |
| | Reweight-DualT max | 65.22±0.28 | 55.01±0.86 | 70.16±0.26 | 56.90±4.55 | 59.62±1.59 | 49.79±0.18 | 65.64±0.71 | 61.73±2.11 |
| | Reweight-DualT 97% | 29.39±0.36 | 29.30±0.74 | 70.24±0.26 | 48.87±6.87 | 25.83±0.16 | 8.51±0.26 | 39.29±0.45 | 35.78±1.10 |
| | Reweight-Ours | **70.10±0.10** | **61.74±0.64** | 70.52±0.26 | **65.78±0.56** | 64.45±0.47 | 58.60±3.30 | **69.40±0.31** | **65.93±0.42** |
| CF1 | Standard | 60.27±0.52 | 22.73±0.15 | 64.66±0.82 | 7.07±0.02 | 62.38±0.27 | 56.78±1.31 | 60.04±0.17 | 45.35±0.60 |
| | GCE | 60.76±0.08 | 21.06±1.12 | 65.27±0.22 | 7.04±0.03 | 63.60±0.25 | 56.72±1.56 | 60.66±0.19 | 44.28±1.12 |
| | CDR | 60.26±0.55 | 22.42±1.78 | 65.27±0.10 | 7.07±0.04 | 62.63±0.08 | 56.41±1.36 | 59.85±0.38 | 45.41±0.85 |
| | AGCN | 61.79±0.98 | 19.30±1.20 | 66.29±0.11 | 7.05±0.04 | 64.09±0.39 | 58.97±1.15 | 60.35±0.54 | 43.89±1.46 |
| | CSRA | 62.46±0.53 | 24.17±2.76 | 65.80±0.02 | 7.06±0.02 | 63.90±0.48 | 55.97±2.29 | 61.30±0.42 | 44.25±2.33 |
| | WSIC | 61.12±0.08 | 27.16±1.54 | 63.71±0.34 | 7.03±0.01 | 61.12±1.30 | 52.47±0.99 | 60.51±0.25 | 45.52±1.32 |
| | Reweight-T max | 61.59±0.51 | 32.78±0.70 | 64.82±0.24 | 56.68±0.80 | 63.35±0.16 | 59.93±0.50 | 60.92±0.08 | 48.40±0.33 |
| | Reweight-T 97% | 55.67±0.45 | 52.79±1.04 | 63.97±0.74 | 56.70±2.11 | 48.47±0.70 | 40.20±0.82 | 55.33±0.12 | 52.19±0.29 |
| | Reweight-DualT max | 64.79±0.23 | 52.16±2.08 | 63.51±0.35 | 54.62±1.12 | 66.29±0.34 | **61.33±0.09** | 65.18±0.22 | 60.88±0.59 |
| | Reweight-DualT 97% | 32.15±0.43 | 30.32±0.91 | 65.23±0.28 | 33.76±8.88 | 29.99±0.10 | 12.77±0.21 | 41.29±0.83 | 37.08±1.29 |
| | Reweight-Ours | **67.18±0.17** | **57.46±0.52** | 65.42±0.49 | **58.63±1.30** | **66.65±0.15** | 61.13±1.01 | **66.42±0.10** | **62.94±0.28** |

## 5 Limitations

This work still has certain limitations, including: 1) This work exploits the memorization effect [4] in deep learning to perform sample selection, while the memorization effect has not been found in other traditional machine learning methods, and therefore, the proposed estimator can not be applied to such learning methods. 2) This work estimates occurrence probabilities using frequency counting. Although this estimation error will converge to zero exponentially fast [6], when the number of one label appearing is too small, e.g., less than 50, the estimation of the transition matrix for this class label is still difficult to be accurate. 3) Since our work assumes label noise is class-dependent but instance-independent, when this assumption does not hold, the estimation is not guaranteed. The discussions about the relaxation of instance-independent assumption can be found in Appendix N, which reveals its applicability in certain typical instance-dependent cases.

## 6 Conclusion

In this paper, we study the estimation problem of the transition matrices in the noisy multi-label setting. We prove some identifiability results of class-dependent transition matrices in such a setting, inspired by which we propose a new estimator to approximate the transition matrix. The proposed estimator utilizes the information of label correlations, and demands neither anchor points nor accurate fitting of noisy class posterior. Experiments on three popular multi-label datasets illustrate the effectiveness of the proposed estimator to accurately estimate transition matrices, and the consistent algorithms with this estimator achieve better classification performance.

## Acknowledgements

Yibing Zhan was partially supported by the Major Science and Technology Innovation 2030 "New Generation Artificial Intelligence" Key Project (No. 2021ZD0111700) and the National Natural Science Foundation of China (No. 62002090). Shiming Ge was partially supported by grants from the Beijing Natural Science Foundation (L192040), and National Natural Science Foundation of China (61772513). Xiaobo Xia was partially supported by Google PhD Fellowship and Australian Research Council Project DE-19010147. Tongliang Liu was partially supported by Australian Research Council Projects DP180103424, DE-190101473, IC-190100031, DP-220102121, and FT-220100318.

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
