## A  Definition of Class-dependent Label Noise

In this paper, the class-dependent multi-label noise for class label $Y^j$ means that the flip probabilities of $\bar{Y}^j$ are only dependent on the value of class label $Y^j$, i.e., $Y^j = 0$ or $Y^j = 1$. And the corresponding class-dependent transition matrix represents $P(\bar{Y}^j|Y^j)$. The differences between this definition and the class-dependent label noise in the single-label cases are as follows:

First, the class-dependent label noise in the single-label cases represents the flip probability from class $i$ to class $j$ ($i$ and $j$ are two different classes), while in this paper, the class-dependent label noise for class label $Y^j$ is only dependent on $Y^j = 0$ or $Y^j = 1$, which is independent on another class label $Y^i$, i.e., $P\left(\bar{Y}^j|Y^j, Y^i\right) = P\left(\bar{Y}^j|Y^j\right)$.

Second, the class-dependent label noise in the single-label cases can be modeled by a $C \times C$ transition matrix, bridging the transition from clean single label to noisy single label, while in this paper, the class-dependent label noise for class label $Y^j$ can be modeled by a $2 \times 2$ transition matrix, bridging the transition from clean label $Y^j$ to noisy label $\bar{Y}^j$.

Third, the definition of class-dependent multi-label noise in this paper can be extended to instance-dependent multi-label noise, where the flip probabilities of $\bar{Y}^j$ are dependent on class label $Y^j$ and instance feature $X$. Such instance-dependent label model can sufficiently model various multi-label noise cases such as missing multi-labels [45, 44], partial multi-labels [53, 55], pair-wise label noise [13, 23, 24], and PMD label noise [59]. While the instance-dependent label noise in the single-label cases can not simultaneously model such complex multi-label noise.

## B  Statistically Consistent Algorithms for Noisy Multi-label Learning

Many statistically consistent algorithms [28, 33, 52, 62] have been proposed for noisy multi-class learning. By decomposing the multi-label task into multiple conditionally independent binary classification problems, we can apply these consistent algorithms to each binary classification problem for noisy multi-label learning. Without loss of generality, we present applying a risk-consistent algorithm, i.e., Reweight [28, 52], in noisy multi-label learning here.

First of all, we decompose the task into q independent binary classification problems given $X$, which is a widely used assumption for deep multi-label learning [7, 36, 54] and the surrogate loss is as:

$$\mathcal{L}(\boldsymbol{f}(\boldsymbol{X}), \boldsymbol{Y}) = \sum_{j=1}^{q} \ell(f_j(\boldsymbol{X}), Y^j) \tag{8}$$

where $\boldsymbol{f} = (f_1, f_2, ..., f_q)$ is the learnable $q$ classification functions, and $\ell$ is the base loss function. In the deep learning commonunity, $\boldsymbol{f}$ is usually modeled by a deep nerual network with the outputs of $q$ sigmoid functions, and $\ell$ is usually the binary cross entropy function.

Similar to the single-label case [28, 52], Reweight method employs the importance reweighting technique to rewrite the expected risk w.r.t. clean data:

$$
\begin{aligned}
R(\boldsymbol{f}) &= \mathbb{E}_{(\boldsymbol{X},\boldsymbol{Y})\sim D}[\mathcal{L}(\boldsymbol{f}(\boldsymbol{X}), \boldsymbol{Y})] = \sum_{j=1}^{q} \mathbb{E}_{(\boldsymbol{X},\boldsymbol{Y})}[\ell(f_j(\boldsymbol{X}), Y^j)] \\
&= \sum_{j=1}^{q} \int_{\boldsymbol{x}} \sum_{i} P_D(\boldsymbol{X} = \boldsymbol{x}, Y^j = i)\ell(f_j(\boldsymbol{x}), i)d\boldsymbol{x} \\
&= \sum_{j=1}^{q} \int_{\boldsymbol{x}} \sum_{i} P_{\bar{D}}(\boldsymbol{X} = \boldsymbol{x}, \bar{Y}^j = i)\frac{P_D(\boldsymbol{X} = \boldsymbol{x}, Y^j = i)}{P_{\bar{D}}(\boldsymbol{X} = \boldsymbol{x}, \bar{Y}^j = i)}\ell(f_j(\boldsymbol{x}), i)d\boldsymbol{x} \\
&= \sum_{j=1}^{q} \int_{\boldsymbol{x}} \sum_{i} P_{\bar{D}}(\boldsymbol{X} = \boldsymbol{x}, \bar{Y}^j = i)\frac{P_D(Y^j = i \mid \boldsymbol{X} = \boldsymbol{x})}{P_{\bar{D}}(\bar{Y}^j = i \mid \boldsymbol{X} = \boldsymbol{x})}\ell(f_j(\boldsymbol{x}), i)d\boldsymbol{x} \\
&= \sum_{j=1}^{q} \mathbb{E}_{(\boldsymbol{X},\bar{\boldsymbol{Y}})}[\bar{\ell}_j(f_j(\boldsymbol{X}), \bar{Y}^j)] = \mathbb{E}_{(\boldsymbol{X},\bar{\boldsymbol{Y}})\sim \bar{D}}[\bar{\mathcal{L}}(\boldsymbol{f}(\boldsymbol{X}), \bar{\boldsymbol{Y}})],
\end{aligned}
\tag{9}
$$

where $D$ denotes the distribution for clean data, $\bar{D}$ for noisy data, $\bar{\ell}_j(f_j(\boldsymbol{x}), i) = \frac{P_D(Y^j=i|\boldsymbol{X}=\boldsymbol{x})}{P_{\bar{D}}(Y^j=i|\boldsymbol{X}=\boldsymbol{x})}\ell(f_j(\boldsymbol{x}), i)$, $\bar{\mathcal{L}}(\boldsymbol{f}(\boldsymbol{x}), \bar{\boldsymbol{y}}) = \sum_{j=1}^{q} \bar{\ell}_j(f_j(\boldsymbol{x}), \bar{Y}^j)$, and the third last equation holds because label noise is assumed to be independent of instances. In the paper, we have omitted the subscript for $P$ when no confusion is caused.

We use the output of the sigmoid function $g_j(\boldsymbol{x})$ to approximate $P(Y^j=1|X=\boldsymbol{x})$, i.e, $P(Y^j=1|X=\boldsymbol{x}) \approx \hat{P}(Y^j=1|X=\boldsymbol{x}) = g_j(\boldsymbol{x})$ and $P(Y^j=0|X=\boldsymbol{x}) \approx \hat{P}(Y^j=0|\boldsymbol{X}=\boldsymbol{x}) = 1 - g_j(\boldsymbol{x})$. Then, $\hat{P}(\bar{Y}^j=k \mid \boldsymbol{X}=\boldsymbol{x}) = \sum_{i=0}^{1} T_{ik}^j \hat{P}(Y^j=i \mid \boldsymbol{X}=\boldsymbol{x})$ is an approximation for $P(\bar{Y}^j=k \mid \boldsymbol{X}=\boldsymbol{x})$. By employing Reweight algorithm, we build the risk-consistent estimator as:

$$\bar{R}_{n,w}(\{T^j\}_{j=1}^{q}, \boldsymbol{f}) = \frac{1}{n}\sum_{i=1}^{n}\sum_{j=1}^{q} \frac{\hat{P}(Y^j=\bar{y}_i^j \mid \boldsymbol{X}=\boldsymbol{x}_i)}{\hat{P}(\bar{Y}^j=\bar{y}_i^j \mid \boldsymbol{X}=\boldsymbol{x}_i)}\ell\left(f_j\left(\boldsymbol{x}_i\right), \bar{y}_i^j\right), \tag{10}$$

where $f_j(\boldsymbol{x}) = \mathbb{I}[g_j(\boldsymbol{x}) > 0.5]$, and $\mathbb{I}[.]$ is the indicator function which takes 1 if the identity index is true and 0 otherwise; the subscript $w$ denotes that the loss function is weighted.

## C    Validation of Assumption 2

Table 7: The frequencies of $Y^i=0$ given $Y^{50}=0/1$ on MS-COCO training dataset. The frequencies whose difference between given $Y^{50}=0$ and $Y^{50}=1$ is greater than 0.015 are in **bold**.

| $i=1$ | $i=2$ | $i=3$ | $i=4$ | $i=5$ | $i=6$ | $i=7$ | $i=8$ | $i=9$ | $i=10$ |
|---|---|---|---|---|---|---|---|---|---|
| **0.960/0.983** | 0.979/0.992 | **0.991/0.920** | 0.974/0.985 | **0.999/0.961** | **0.999/0.959** | **0.984/0.999** | **0.955/0.980** | **0.974/0.936** | **0.989/0.958** |
| $i=11$ | $i=12$ | $i=13$ | $i=14$ | $i=15$ | $i=16$ | $i=17$ | $i=18$ | $i=19$ | $i=20$ |
| **0.955/0.987** | 0.980/0.970 | **0.943/0.964** | 0.925/0.929 | **0.920/0.954** | **0.969/0.996** | **0.983/0.952** | 0.975/0.975 | **0.931/0.866** | **0.975/0.995** |
| $i=21$ | $i=22$ | $i=23$ | $i=24$ | $i=25$ | $i=26$ | $i=27$ | $i=28$ | $i=29$ | $i=30$ |
| **0.937/0.989** | **0.982/0.941** | **0.912/0.874** | **0.950/0.971** | 0.958/0.964 | **0.974/0.990** | 0.918/0.923 | **0.886/0.908** | 0.957/0.968 | 0.987/0.987 |
| $i=31$ | $i=32$ | $i=33$ | $i=34$ | $i=35$ | $i=36$ | $i=37$ | $i=38$ | $i=39$ | $i=40$ |
| 0.976/0.986 | 0.981/0.989 | 0.966/0.972 | **0.994/0.972** | **0.959/0.993** | 0.998/0.999 | **0.989/0.902** | 0.982/0.969 | 0.990/0.990 | **0.972/0.991** |
| $i=41$ | $i=42$ | $i=43$ | $i=44$ | $i=45$ | $i=46$ | $i=47$ | $i=48$ | $i=49$ | $i=50$ |
| **0.997/0.967** | 0.962/0.962 | 0.966/0.973 | 0.980/0.992 | **0.987/0.957** | **0.974/0.993** | **0.977/0.992** | 0.965/0.984 | 0.993/0.995 | **1.000/0.000** |
| $i=51$ | $i=52$ | $i=53$ | $i=54$ | $i=55$ | $i=56$ | $i=57$ | $i=58$ | $i=59$ | $i=60$ |
| 0.970/0.975 | **0.954/0.969** | 0.972/0.986 | 0.981/0.967 | 0.974/0.985 | 0.989/0.994 | **0.977/0.994** | **0.930/0.984** | **0.999/0.945** | **0.999/0.952** |
| $i=61$ | $i=62$ | $i=63$ | $i=64$ | $i=65$ | $i=66$ | $i=67$ | $i=68$ | $i=69$ | $i=70$ |
| **0.999/0.975** | 0.965/0.973 | **0.997/0.936** | 0.980/0.990 | 0.984/0.977 | **0.998/0.950** | 0.976/0.987 | **0.999/0.948** | **0.997/0.944** | 0.997/0.999 |
| $i=71$ | $i=72$ | $i=73$ | $i=74$ | $i=75$ | $i=76$ | $i=77$ | $i=78$ | $i=79$ | $i=80$ |
| **0.945/0.994** | 0.991/0.992 | 0.969/0.961 | 0.964/0.975 | **0.959/0.938** | **0.950/0.970** | **0.991/0.947** | **0.948/0.987** | 0.984/0.974 | **0.966/0.998** |

In order to verify the assumption 2, we count the frequencies of $Y^i=0$ given $Y^{50}=0/1$ on MS-COCO training dataset, i.e., $\hat{P}(Y^i=0|Y^{50}=0)/\hat{P}(Y^i=0|Y^{50}=1)$. According to Hoeffding's inequality [6], when the frequencies whose difference between given $Y^{50}=0$ and $Y^{50}=1$ is greater than 0.015, we have at least 95% confidence to make sure $P(Y^i=0|Y^{50}=0) \neq P(Y^i=0|Y^{50}=1)$. As shown in Tab. 7, class 50 have the correlation with another 46 classes with a high probability, which means they are very likely to hold Assumption 2, accounting for the majority of all 79 classes.

In the implementation of our estimator, we always assume class $j$ and class $i$ have a strong correlation at first, and use them to estimate. While, when a reasonable solution cannot be obtained, we will abandon class $i$ and choose another class.

## D    Discussion about Significantly Different Label Pairs in Real-world Datasets

Although the real-world scenarios have complex label correlations, we claim significantly different class label pairs usually account for the majority of all label pairs in the typical multi-label datasets, e.g., MS-COCO [27] and OpenImages [19] datasets. It is because among a large number of classes, most label pairs belong to significantly different superclasses. For example, in MS-COCO datasets, there are 80 classes, which belong to 10 significantly different superclasses (outdoor, food, indoor, appliance, sports, person, animal, vehicle, furniture, accessory, electronic, kitchen). In OpenImages dataset, there are 19,957 class labels, and also have significantly different superclasses, such as Toy, Budilding, Medical equipment, Clothing, Insect and so on. A part of these superclasses can be seen in [1], which clearly shows most of the label pairs are significantly different and do not share the major discriminant features.

# E  Proof of Theorem 1

**Lemma 1.** *$\bar{Y}^i$ and $\bar{Y}^j$ are independent given $Y^j$.*

*Proof.* Since we assume that the transition matrix is class-dependent and instance-independent, $\bar{Y}^j$ and any variable are independent given $Y^j$. Therefore, this lemma holds. $\qquad\square$

**Lemma 2.** *The product of two row-stochastic matrices is still a row-stochastic one.*

*Proof.* Let $P$ and $Q$ be row-stochastic matrices of the following form: $P = \begin{pmatrix} 1-p_- & p_- \\ p_+ & 1-p_+ \end{pmatrix}$ and $Q = \begin{pmatrix} 1-q_- & q_- \\ q_+ & 1-q_+ \end{pmatrix}$. Then the product of $P$ and $Q$ is $PQ = \begin{pmatrix} 1-q_-+p_-(-1+q_-+q_+) & q_--p_-(-1+q_-+q_+) \\ q_+-p_+(-1+q_-+q_+) & 1-q_++p_+(-1+q_-+q_+) \end{pmatrix}$. It can be readily verified that the sum of each row of $PQ$ is equal to 1, meaning that $PQ$ is a row-stochastic matrix. $\qquad\square$

**Theorem 1.** *Two noisy labels $\{\bar{Y}^j, \bar{Y}^i\}$ will not suffice to identify $T^j$.*

*Proof.* First, the information from $\bar{Y}^j, \bar{Y}^i$ can be fully captured by the following four quantities: $P\left(\bar{Y}^j = 0, \bar{Y}^i = 0\right)$, $P\left(\bar{Y}^j = 1, \bar{Y}^i = 0\right)$, $P\left(\bar{Y}^j = 0, \bar{Y}^i = 1\right)$, and $P\left(\bar{Y}^j = 1, \bar{Y}^i = 1\right)$. According to Lemma 1, these four quantities can lead to four equations that depend on $T^j$:

$$P\left(\bar{Y}^j = 0, \bar{Y}^i = 0\right) = P(Y^j=0)T_{00}^j P(\bar{Y}^i=0|Y^j=0) + P(Y^j=1)T_{10}^j P(\bar{Y}^i=0|Y^j=1)$$
$$P\left(\bar{Y}^j = 0, \bar{Y}^i = 1\right) = P(Y^j=0)T_{00}^j P(\bar{Y}^i=1|Y^j=0) + P(Y^j=1)T_{10}^j P(\bar{Y}^i=1|Y^j=1)$$
$$P\left(\bar{Y}^j = 1, \bar{Y}^i = 0\right) = P(Y^j=0)T_{01}^j P(\bar{Y}^i=0|Y^j=0) + P(Y^j=1)T_{11}^j P(\bar{Y}^i=0|Y^j=1)$$
$$P\left(\bar{Y}^j = 1, \bar{Y}^i = 1\right) = P(Y^j=0)T_{01}^j P(\bar{Y}^i=1|Y^j=0) + P(Y^j=1)T_{11}^j P(\bar{Y}^i=1|Y^j=1).$$

For simplicity, we denote

$$E = \begin{pmatrix} P(\bar{Y}^j=0, \bar{Y}^i=0) & P(\bar{Y}^j=0, \bar{Y}^i=1) \\ P(\bar{Y}^j=1, \bar{Y}^i=0) & P(\bar{Y}^j=1, \bar{Y}^i=1) \end{pmatrix} = \begin{pmatrix} e_{00} & e_{01} \\ e_{10} & e_{11} \end{pmatrix},$$

$$P = \begin{pmatrix} P(Y^j=0) & 0 \\ 0 & P(Y^j=1) \end{pmatrix} = \begin{pmatrix} 1-p & 0 \\ 0 & p \end{pmatrix},$$

$$T^j = \begin{pmatrix} P(\bar{Y}^j=0 \mid Y^j=0) & P(\bar{Y}^j=1 \mid Y^j=0) \\ P(\bar{Y}^j=0 \mid Y^j=1) & P(\bar{Y}^j=1 \mid Y^j=1) \end{pmatrix} = \begin{pmatrix} 1-\rho_- & \rho_- \\ \rho_+ & 1-\rho_+ \end{pmatrix}, \text{ and}$$

$$M = \begin{pmatrix} P(\bar{Y}^i=0 \mid Y^j=0) & P(\bar{Y}^i=1 \mid Y^j=0) \\ P(\bar{Y}^i=0 \mid Y^j=1) & P(\bar{Y}^i=1 \mid Y^j=1) \end{pmatrix} = \begin{pmatrix} 1-\rho'_- & \rho'_- \\ \rho'_+ & 1-\rho'_+ \end{pmatrix}.$$

Then, the system of equations can be expressed as $E = (T^j)^\top P M$, i.e.,

$$\begin{pmatrix} e_{00} & e_{01} \\ e_{10} & e_{11} \end{pmatrix} = \begin{pmatrix} 1-\rho_- & \rho_- \\ \rho_+ & 1-\rho_+ \end{pmatrix}^\top \begin{pmatrix} 1-p & 0 \\ 0 & p \end{pmatrix} \begin{pmatrix} 1-\rho'_- & \rho'_- \\ \rho'_+ & 1-\rho'_+ \end{pmatrix}. \tag{11}$$

Assuming $\{T^0, P^0, M^0\}$ satisfies

$$E = (T^0)^\top P^0 M^0, \tag{12}$$

Next, we will prove that by selecting proper parameters, a different solution of Eq. (11) can be derived, which ruins the identifiability of $T^j$. Let $A = \begin{pmatrix} 1-a_- & a_- \\ a_+ & 1-a_+ \end{pmatrix}$ and $B = \begin{pmatrix} 1-b_- & b_- \\ b_+ & 1-b_+ \end{pmatrix}$ be invertible, row-stochastic matrices. Based on the invertibility of $A$ and $B$, Eq. (12) can be rewritten as $E = (AT^0)^\top (A^\top)^{-1} P^0 B^{-1} B M^0$. According to Lemma 2, $T^1 = AT^0$ and $M^1 = BM^0$ are row-stochastic matrices, which is consistent with the form of $T^j$ and $M$.

Last, denoting $\boldsymbol{P}^0 = \begin{pmatrix} 1-p_0 & 0 \\ 0 & p_0 \end{pmatrix}$ and $\boldsymbol{P}^1 = \begin{pmatrix} p_{00} & p_{01} \\ p_{10} & p_{11} \end{pmatrix}$, by letting $\boldsymbol{P}^1 = (\boldsymbol{A}^\top)^{-1}\boldsymbol{P}^0\boldsymbol{B}^{-1} =$

$$\left[ \begin{pmatrix} 1-a_- & a_- \\ a_+ & 1-a_+ \end{pmatrix}^\top \right]^{-1} \begin{pmatrix} 1-p_0 & 0 \\ 0 & p_0 \end{pmatrix} \begin{pmatrix} 1-b_- & b_- \\ b_+ & 1-b_+ \end{pmatrix} = \begin{pmatrix} p_{00} & p_{01} \\ p_{10} & p_{11} \end{pmatrix} \text{ be in the form of}$$

$\tilde{\boldsymbol{P}}$, i.e., solving the following equations:

$$\begin{cases} p_{00} + p_{11} = 1, \\ p_{01} = 0, \\ p_{10} = 0, \end{cases}$$

we can get $a_+ = \frac{b_-(1-p_0)}{b_- - p_0}$ and $b_+ = \frac{a_-(1-p_0)}{a_- - p_0}$. It means that when we have a solution $\{\boldsymbol{T}^0, \boldsymbol{P}^0, \boldsymbol{M}^0\}$ of Eq. (11), we can get another different solution $\{\boldsymbol{T}^1, \boldsymbol{P}^1, \boldsymbol{M}^1\}$ by setting appropriate values of $b_-$ and $a_-$. Hence, $\boldsymbol{T}^j$ is unidentifiable in this situation.

$\square$

## F   Proof of Theorem 2

To make the proof clear, following [30], we reproduce the Kruskal's identifiability result here. The setup of Kruskal's identifiability result is as follows: suppose that there is an unobserved variable $Z$ that takes values in $\{0, 1, ..., K-1\}$. $Z$ has a non-degenerate prior $P(Z = i) > 0$. Instead of observing $Z$, we observe a set of conditionally independent variables $\{O^{(t)}\}_{t=1}^N$. Each $O^{(t)}$ has a finite state space with cardinality $\kappa_t$. Let $\boldsymbol{M}^{(t)}$ be a matrix of size $K \times \kappa_t$, which $j$-th row is simply $[P(O^{(t)} = 1 \mid Z = j), \ldots, P(O^{(t)} = \kappa_t \mid Z = j)]$. The previous works [21, 40] have proved the following Theorem 5.

**Definition 3** (Kruskal rank [30]). *For a matrix $\boldsymbol{M}$, the Kruskal rank of $\boldsymbol{M}$ is the largest number $I$ such that every set of $I$ rows of $\boldsymbol{M}$ are independent. The symbol is $\mathrm{Kr}(\boldsymbol{M}) = I$.*

**Theorem 5** (Kruskal's identifiability result [21, 40]). *The model parameters are uniquely identifiable, up to label permutation, if*

$$\sum_{t=1}^N \mathrm{Kr}\left(\boldsymbol{M}^{(t)}\right) \geq 2K + N - 1.$$

We can prove Theorem 2 with the following lemmas:

**Lemma 3.** $\bar{Y}^i$ *and* $Y^j$ *are independent given* $Y^i$.

*Proof.* Since we assume that the transition matrix is class-dependent and instance-independent, $\bar{Y}^i$ and any variable are independent given $Y^i$. Therefore, this lemma holds. $\square$

**Lemma 4.** *If* $Y^j \in \{0, 1\}$ *corresponds to* $Z$, $\{\bar{Y}^j, \bar{Y}^i, \bar{Y}^k\}$ *correspond to the observations* $\{O^{(t)}\}_{t=1}^3$, *then* $\mathrm{Kr}\left(\boldsymbol{M}^{(t)}\right) = 2, t \in [3]$.

*Proof.* As $\boldsymbol{M}^{(1)} = \boldsymbol{T}^j$ is a row-stochastic matrix, according to Assumption 1, every set of 2 rows of it are independent, thus its Kruskal rank is 2. Therefore, according to Definition 3, $\mathrm{Kr}\left(\boldsymbol{T}^j\right) = 2$.

The $(p, q)$ entry of $\boldsymbol{M}^{(2)}$ is

$$\begin{aligned} M_{pq}^{(2)} = P(\bar{Y}^i = q \mid Y^j = p) &= \sum_{c=0}^1 P(\bar{Y}^i = q, Y^i = c \mid Y^j = p) \\ &= \sum_{c=0}^1 P(\bar{Y}^i = q \mid Y^i = c, Y^j = p)P(Y^i = c \mid Y^j = p) \qquad (13) \\ &= \sum_{c=0}^1 P(\bar{Y}^i = q \mid Y^i = c)P(Y^i = c \mid Y^j = p), \end{aligned}$$

where the last equation holds because of Lemma. 3.

Denote $\boldsymbol{M}'^{(2)} = \begin{pmatrix} P(Y^i = 0 \mid Y^j = 0) & P(Y^i = 1 \mid Y^j = 0) \\ P(Y^i = 0 \mid Y^j = 1) & P(Y^i = 1 \mid Y^j = 1) \end{pmatrix}$. Then Eq. (13) can be rewritten as the following matrix form:

$$\boldsymbol{M}^{(2)} = \boldsymbol{M}'^{(2)}\boldsymbol{T}^i, \tag{14}$$

where $\boldsymbol{T}^i$ denotes the transition matrix of class $i$. According to Assumption 1, $\mathrm{Kr}\left(\boldsymbol{T}^i\right) = 2$. As $\boldsymbol{M}'^{(2)}$ is a row-stochastic matrix, according to Assumption 2, every set of 2 rows of it are independent. Hence, according to Definition 3, $\mathrm{Kr}\left(\boldsymbol{M}'^{(2)}\right) = 2$.

Since $\boldsymbol{T}^i$ and $\boldsymbol{M}'^{(2)}$ are full-rank matrices, based on Eq. (13), we can get the Kruskal rank of $\boldsymbol{M}^{(2)}$ as $\mathrm{Kr}\left(\boldsymbol{M}^{(2)}\right) = 2$. Similarly, $\mathrm{Kr}\left(\boldsymbol{M}^{(3)}\right) = 2$. □

**Theorem 2.** *If $\bar{Y}^i$ and $\bar{Y}^k$ are independent given $Y^j$, three noisy labels $\{\bar{Y}^j, \bar{Y}^i, \bar{Y}^k\}$ are sufficient to identify $\boldsymbol{T}^j$.*

*Proof.* According to Lemma 1 and that $\bar{Y}^i$ and $\bar{Y}^k$ are independent given $Y^j$, we can relate our multi-label noise setting to the setup of Kruscal's identifiability scenario: $Y^j \in \{0, 1\}$ corresponds to the unobserved hidden variable $Z$; $P(Y^j = i)$ corresponds to the prior of this hidden variable; Noisy labels $\{\bar{Y}^j, \bar{Y}^i, \bar{Y}^k\}$ correspond to the observations $\{O^{(t)}\}_{t=1}^3$. $\kappa_t$ is then simply the cardinality of the noisy label space, i.e., $\kappa_t = K = 2$; Each $O^{(t)}$ has a corresponding observation matrix $\boldsymbol{M}^{(t)}$, and $\boldsymbol{M}^{(t)}_{vk} = P\left(O^{(t)} = k \mid Y^j = v\right)$. Now we can get the following result about identifiability of $\boldsymbol{T}$.

According to Lemma 4, the Kruskal ranks satisfy

$$\sum_{t=1}^3 \mathrm{Kr}\left(\boldsymbol{M}^{(t)}\right) = 3K = 2K + 2 \geq 2K + N - 1, \text{ when } N = 3.$$

Calling Theorem 5 proves the uniqueness of $\boldsymbol{M}^{(t)}$. As $\boldsymbol{M}^{(1)} = \boldsymbol{T}^j$, then $\boldsymbol{T}^j$ is identifiable. □

## G  Proof of Theorem 3

**Lemma 5.** *$\bar{Y}^i, \bar{Y}^j$ and $\bar{Y}^k$ are independent given $Y^j$ and $Y^i$.*

*Proof.* Since we assume that the transition matrix is class-dependent and instance-independent, $\bar{Y}^j$ and any variable are independent given $Y^j$, and $\bar{Y}^i$ and any variable are independent given $Y^i$. Therefore, this lemma holds. □

**Lemma 6.** *If $\boldsymbol{M}_{4\times 2} = P(\bar{Y}^i \mid Y^j, Y^i)$, the first two rows and last two rows of $\boldsymbol{M}$ are identical respectively, i.e., $M_{0p} = M_{1p}$ and $M_{2p} = M_{3p}, p = 0, 1$.*

*Proof.*

$$\boldsymbol{M} = \begin{pmatrix} P(\bar{Y}^i = 0 \mid Y^j = 0, Y^i = 0) & P(\bar{Y}^i = 1 \mid Y^j = 0, Y^i = 0) \\ P(\bar{Y}^i = 0 \mid Y^j = 1, Y^i = 0) & P(\bar{Y}^i = 1 \mid Y^j = 1, Y^i = 0) \\ P(\bar{Y}^i = 0 \mid Y^j = 0, Y^i = 1) & P(\bar{Y}^i = 1 \mid Y^j = 0, Y^i = 1) \\ P(\bar{Y}^i = 0 \mid Y^j = 1, Y^i = 1) & P(\bar{Y}^i = 1 \mid Y^j = 1, Y^i = 1) \end{pmatrix}$$

$$= \begin{pmatrix} P(\bar{Y}^i = 0 \mid Y^i = 0) & P(\bar{Y}^i = 1 \mid Y^i = 0) \\ P(\bar{Y}^i = 0 \mid Y^i = 0) & P(\bar{Y}^i = 1 \mid Y^i = 0) \\ P(\bar{Y}^i = 0 \mid Y^i = 1) & P(\bar{Y}^i = 1 \mid Y^j = 1) \\ P(\bar{Y}^i = 0 \mid Y^j = 1) & P(\bar{Y}^i = 1 \mid Y^j = 1) \end{pmatrix},$$

where the second equation holds because $\bar{Y}^i$ is only dependent on $Y^i$. Accordingly, $M_{0p} = M_{1p}$ and $M_{2p} = M_{3p}, p = 0, 1$. □

**Theorem 3.** *If $\bar{Y}^i$ and $\bar{Y}^k$ are not independent given $Y^j$, three noisy labels $\{\bar{Y}^j, \bar{Y}^i, \bar{Y}^k\}$ will not suffice to identify $\boldsymbol{T}^j$.*

*Proof.* The likelihood of noisy labels can be formulated as a third-order tensor $\boldsymbol{L}$, where the $(p, q, m)$ entry is

$$L_{pqm} = P(\bar{Y}^i = p, \bar{Y}^j = q, \bar{Y}^k = m). \tag{15}$$

According to Lemma 5, Eq. (15) can be expanded by Bayes Rule:

$$
\begin{aligned}
L_{pqm} &= \sum_{c_0,c_1=0}^{c_0,c_1=1} P(\bar{Y}^i = p, \bar{Y}^j = q, \bar{Y}^k = m \mid Y^j = c_0, Y^i = c_1) P(Y^j = c_0, Y^i = c_1) \\
&= \sum_{c_0,c_1=0}^{c_0,c_1=1} P(\bar{Y}^i = p \mid Y^j, Y^i) P(\bar{Y}^j = q \mid Y^j, Y^i) P(\bar{Y}^k = m \mid Y^j, Y^i) P(Y^j, Y^i).
\end{aligned}
\tag{16}
$$

where we omit the value of $Y^i, Y^j$ in the last equation for simplicity. Let $\boldsymbol{M}^{(1)}_{4\times2} = P(\bar{Y}^j \mid Y^i, Y^j)$, $\boldsymbol{M}^{(2)}_{4\times2} = P(\bar{Y}^i \mid Y^i, Y^j)$, $\boldsymbol{M}^{(3)}_{4\times2} = P(\bar{Y}^k \mid Y^i, sY^j)$ and $\boldsymbol{\Lambda}_{4\times1} = P(Y^i, Y^j)$, and the Eq. (16) can be expressed as $L_{pqm} = \sum_{v=0}^{3} M^{(2)}_{vp} M^{(1)}_{vq} M^{(3)}_{vm} \Lambda_v$.

Let $\{\boldsymbol{A}^0, \boldsymbol{B}^0, \boldsymbol{C}^0, \boldsymbol{\Lambda}^0\}$ be a solution of $\{\boldsymbol{M}^{(1)}, \boldsymbol{M}^{(2)}, \boldsymbol{M}^{(3)}, \boldsymbol{\Lambda}\}$, which means it fulfils the likelihood equations (Eq. (16)), i.e.,

$$L_{pqm} = \sum_{v=0}^{3} B^0_{vp} A^0_{vq} C^0_{vm} \Lambda^0_v. \tag{17}$$

Note that as stated in Lemma. 6, the first two rows and last two rows of $\boldsymbol{B}^0$ are identical respectively. Next, we will show that a different solution can be constructed by simply switching the corresponding rows in $\boldsymbol{A}^0$, $\boldsymbol{C}^0$ and $\boldsymbol{\Lambda}^0$, which is consistent with the result in [20, 42] when $\mathrm{Kr}(\boldsymbol{M}^{(2)}) = 1$. By letting

$$
\boldsymbol{A}^1 = \begin{pmatrix} A^0_{10} & A^0_{11} \\ A^0_{00} & A^0_{01} \\ A^0_{30} & A^0_{31} \\ A^0_{20} & A^0_{21} \end{pmatrix}, \quad
\boldsymbol{C}^1 = \begin{pmatrix} C^0_{10} & C^0_{11} \\ C^0_{00} & C^0_{01} \\ C^0_{30} & C^0_{31} \\ C^0_{20} & C^0_{21} \end{pmatrix}, \quad \text{and} \quad
\boldsymbol{\Lambda}^1 = \begin{pmatrix} \Lambda^0_1 \\ \Lambda^0_0 \\ \Lambda^0_3 \\ \Lambda^0_2 \end{pmatrix},
$$

then the Eq. (17) is equivalent to

$$
\begin{aligned}
L_{pqm} &= B^0_{0p} A^0_{0q} C^0_{0m} \Lambda^0_0 + B^0_{1p} A^0_{1q} C^0_{1m} \Lambda^0_1 + B^0_{2p} A^0_{2q} C^0_{2m} \Lambda^0_2 + B^0_{3p} A^0_{3q} C^0_{3m} \Lambda^0_3 \\
&= B^0_{1p} A^0_{0q} C^0_{0m} \Lambda^0_0 + B^0_{0p} A^0_{1q} C^0_{1m} \Lambda^0_1 + B^0_{3p} A^0_{2q} C^0_{2m} \Lambda^0_2 + B^0_{2p} A^0_{3q} C^0_{3m} \Lambda^0_3 \\
&= B^0_{1p} A^1_{1q} C^1_{1m} \Lambda^1_1 + B^0_{0p} A^1_{0q} C^1_{0m} \Lambda^1_0 + B^0_{3p} A^1_{3q} C^1_{3m} \Lambda^1_3 + B^0_{2p} A^1_{2q} C^1_{2m} \Lambda^1_2 \\
&= \sum_{v=0}^{3} B^0_{vp} A^1_{vq} C^1_{vm} \Lambda^1_v
\end{aligned}
\tag{18}
$$

where the second equation holds because the first two rows and last two rows of $\boldsymbol{B}^0$ are identical respectively, i.e., $B^0_{0p} = B^0_{1p}$ and $B^0_{2p} = B^0_{3p}$. Note that $\boldsymbol{A}^1, \boldsymbol{C}^1$ and $\boldsymbol{\Lambda}^1$ are consistent with the form of $\boldsymbol{M}^{(1)}, \boldsymbol{M}^{(3)}$ and $\boldsymbol{\Lambda}$ respectively. Then, according to Eq. (18), it can be readily observed that the $\{\boldsymbol{A}^1, \boldsymbol{B}^0, \boldsymbol{C}^1, \boldsymbol{\Lambda}^1\}$ is a new solution of $\{\boldsymbol{M}^{(1)}, \boldsymbol{M}^{(2)}, \boldsymbol{M}^{(3)}, \boldsymbol{\Lambda}\}$, hence the uniqueness is not guaranteed under this circumstance. As $\boldsymbol{M}^{(1)} = \begin{pmatrix} \boldsymbol{T}^j \\ \boldsymbol{T}^j \end{pmatrix}$ is not unique, then $\boldsymbol{T}^j$ is unidentifiable. $\quad\square$

Note that according to Definition 2, in the situation of Theorem 3, the model parameter $\theta := \{\boldsymbol{T}^j, P(Y^j), P(\bar{Y}^i, \bar{Y}^k | Y^j)\}$. For convenience, in the proof of Theorem 3, we use $\theta^1 := \{\boldsymbol{T}^j, P(Y^j, Y^i), P(\bar{Y}^i | Y^j, Y^i), P(\bar{Y}^k | Y^j, Y^i)\}$ as the model parameter. It is easy to prove that the above different solutions of $\theta^1$ can lead to two different solutions of $\theta$.

## H  Proof of Theorem 4

**Theorem 4.** *If $P(\bar{Y}^i \mid Y^j)$ is known, two noisy labels $\{\bar{Y}^j, \bar{Y}^i\}$ are sufficient to identify $\boldsymbol{T}^j$.*

*Proof.* The proof of Theorem 4 is much similar to that of Theorem 1, we can get a system of equations expressed as $\boldsymbol{E} = (\boldsymbol{T}^j)^\top \boldsymbol{P} \boldsymbol{M}$. The difference lies in that in Theorem 4, the matrix $\boldsymbol{M}$ which is parameterized by $P(\bar{Y}^i \mid Y^j)$ is given. Since $\boldsymbol{M}$ is invertible (Similar to Lemma. 4), the problem can be converted to a simple bilinear decomposition problem:

$$\boldsymbol{E}(\boldsymbol{M})^{-1} = (\boldsymbol{T}^j)^\top \boldsymbol{P},$$

i.e.,

$$\begin{pmatrix} e_{00} & e_{01} \\ e_{10} & e_{11} \end{pmatrix} \begin{pmatrix} 1-\rho'_- & \rho'_- \\ \rho'_+ & 1-\rho'_+ \end{pmatrix}^{-1} = \begin{pmatrix} 1-\rho_- & \rho_- \\ \rho_+ & 1-\rho_+ \end{pmatrix}^\top \begin{pmatrix} 1-p & 0 \\ 0 & p \end{pmatrix}. \tag{19}$$

Solving the above bilinear decomposition problem, the unique solution can be obtained as:

$$p = \frac{(1-\rho'_-) - (e_{00} + e_{10})}{1 - \rho'_- - \rho'_+}. \tag{20}$$

Substituting Eq. (20) into Eq. (19), then right multiplying $\boldsymbol{P}^{-1}$ on both side of the equation, the matrix $\boldsymbol{T}^j$ can be derived as:

$$\boldsymbol{T}^j = [\boldsymbol{E}(\boldsymbol{M})^{-1}(\boldsymbol{P})^{-1}]^\top,$$

which indicates that $\boldsymbol{T}^j$ is identifiable given label correlation $P(\bar{Y}^i \mid Y^j)$. $\qquad\square$

# I  Summary of the Inspirations from the proof of Theorem 1-4

From the proof of Theorem 1, we can know that the label correlations of two noisy labels $\{\bar{Y}^j, \bar{Y}^i\}$ can not offer enough information to achieve the identifiability of $\boldsymbol{T}^j$.

From the proof of Theorem 2, three noisy labels $\{\bar{Y}^j, \bar{Y}^i, \bar{Y}^k\}$ can provide more information than two noisy labels $\{\bar{Y}^j, \bar{Y}^i, \bar{Y}^k\}$ to achieve the identifiability when $\bar{Y}^i$ and $\bar{Y}^k$ are independent given $Y^j$. Note that when satisfying the condition, the parameter $P(\bar{Y}^i, \bar{Y}^k|Y^j)$ is reduced to $P(\bar{Y}^i|Y^j)$ and $P(\bar{Y}^k|Y^j)$.

From the proof of Theorem 3, due to the entangled correlations, when $\bar{Y}^i$ and $\bar{Y}^k$ are not independent given $Y^j$, modelling three noisy labels $\{\bar{Y}^j, \bar{Y}^i, \bar{Y}^k\}$ will increase too many model parameters, making the identifiability decrease from the situation in Theorem 2.

From the proof of Theorem 4, by reducing the unknown model parameters via some extra information, the label correlations of two noisy labels $\{\bar{Y}^j, \bar{Y}^i\}$ can achieve the identifiability.

The summary of $\Omega$, $\theta$, $P_\theta$, condition and the identifiability in Theorem 1-4 is shown in Tab. 8.

Table 8: Summary of $\Omega$, $\theta$, $P_\theta$, condition and the identifiability in Theorem 1-4 .

| | $\Omega$ | $\theta$ | $P_\theta$ | Condition | Identifiability of $\boldsymbol{T}^j$ |
|---|---|---|---|---|---|
| Theorem 1 | $\bar{Y}^j, \bar{Y}^i$ | $T^j, P(Y^j), P(\bar{Y}^i|Y^j)$ | $P(\bar{Y}^j, \bar{Y}^i)$ | – | |
| Theorem 2 | $\bar{Y}^j, \bar{Y}^i, \bar{Y}^k$ | $T^j, P(Y^j), P(\bar{Y}^i|Y^j), P(\bar{Y}^k|Y^j)$ | $P(\bar{Y}^j, \bar{Y}^i, \bar{Y}^k)$ | $\bar{Y}^i$ and $\bar{Y}^k$ are independent given $Y^j$ | ✓ |
| Theorem 3 | $\bar{Y}^j, \bar{Y}^i, \bar{Y}^k$ | $T^j, P(Y^j), P(\bar{Y}^i, \bar{Y}^k|Y^j)$ | $P(\bar{Y}^j, \bar{Y}^i, \bar{Y}^k)$ | – | |
| Theorem 4 | $\bar{Y}^j, \bar{Y}^i$ | $T^j, P(Y^j)$ | $P(\bar{Y}^j, \bar{Y}^i)$ | $P(\bar{Y}^i \mid Y^j)$ is known | ✓ |

# J  Ablation Study about Sample Selection Threshold

The sample selection we adopted [38, 22] is to estimate this clean probability of examples by modeling loss values with a GMM model using the Expectation-Maximization algorithm. If the clean sample can be distinguished according to loss values, and its estimated probability is accurate, the best threshold will be about 0.5. Hence, $\tau = 0.5$ is a typical value in related works [38, 22], and we follow this practice in our experiments. In this section, we conduct the ablation study about the sample selection threshold experiments on noisy VOC2007 datasets. As shown in Fig. 1, $\tau = 0.5$ is a good choice both according to mAP scores on the noisy validation set and according to mAP scores on the clean test dataset.

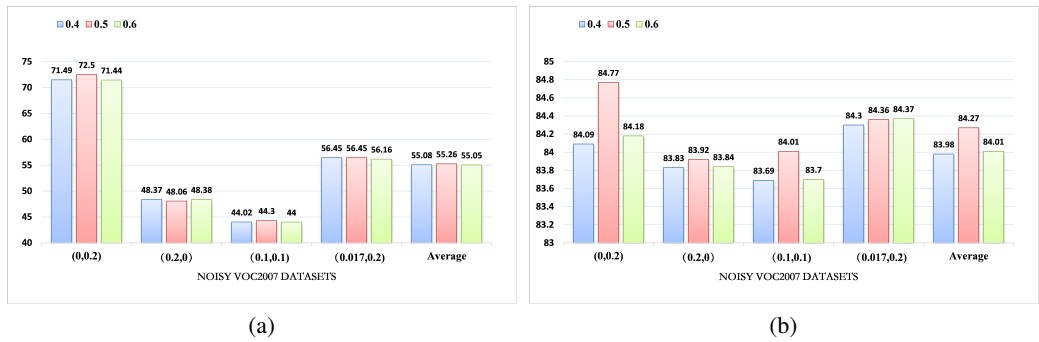

(a)                                    (b)

Figure 1: (a) mAP scores on the noisy validation set of Reweight methods with our estimator with $\tau = 0.4, 0.5, 0.6$; (b) mAP scores on the clean test dataset of Reweight methods with our estimator with $\tau = 0.4, 0.5, 0.6$.

## K   Ablation Study about Label Correlations

According to the results in Appendix C, we divide all labels on MS-COCO dataset into two categories: one is very likely to has a correlation with class label 50 (the label belonging to this category is termed as "label A") and the other is less likely to has a correlation with class label 50 (the label belonging to this category is termed as "label B"). Tab. 9 shows the mean estimation error of transition matrices for class label 50 by using the correlations of label 50 and label A (or B) in various cases. The results represent using the correlations with label A, the estimation error of transition matrices $T^{49}$ will be smaller, showing stronger label correlations can lead to better estimation in our approach.

Table 9: Mean estimation error of transition matrices $T^{49}$ for class label 50 by using the correlations of label 50 and label A (or B) on MS-COCO dataset. The best results are in **bold**.

| Noise rates $(\rho_-, \rho_+)$ | (0,0.2) | (0,0.6) | (0.2,0) | (0.6,0) | (0.1,0.1) | (0.2,0.2) | (0.017,0.2) | (0.034,0.4) |
|---|---|---|---|---|---|---|---|---|
| Using the correlations with label A | **0.02±0.02** | **0.02±0.02** | **0.09±0.04** | **0.07±0.06** | **0.10±0.05** | **0.10±0.08** | **0.09±0.04** | **0.08±0.03** |
| Using the correlations with label B | 0.05±0.06 | 0.05±0.07 | 0.13±0.11 | 0.15±0.14 | 0.19±0.12 | 0.18±0.11 | 0.17±0.10 | 0.17±0.11 |

## L   Comparison with Different Loss Correction Ways

Many statistically consistent algorithms [28, 33, 52] consist of a two-step training procedure. The first step estimates the transition matrix, and the second step builds statistically consistent algorithms via modifying loss functions. Our proposed estimator can be seamlessly embedded into their frameworks. In this section, we compare classification performance of applying our estimation method to four different loss correction ways: Reweight [28] (named Reweight-Ours), Backward [32, 33, 54] (named Backward-Ours), Forward [33] (named Forward-Ours) and T-Revision [52] (named Revision-Ours) on Pascal-VOC2007 dataset. As shown in Tab. 10, T-Revision with our estimation achieves the best performance in most cases, which may be because it can further tune the transition matrices via classification learning.

## M   Comparison with Different Base Learning Algorithms

Recently, many advanced multi-label learning algorithms [58, 61, 58, 5], which designed exquisite networks for multi-label learning, have been proposed, and they perform well on the clean dataset. As their loss functions are similar to Eq. (8), we can also apply reweight method with our estimator to their frameworks. In this section, we compare the classification performance of applying Reweight method with our estimator to three different base learning algorithms: Standard (named "Reweight-Ours"), AGCN [58] (named "AGCN-R-Ours") and CSRA [61] (named "CSRA-R-Ours") on Pascal-VOC2007 dataset. Besides, to show the importance of the consistent algorithms, we also present the performance of base learning algorithms here. As shown in Tab. 11, the consistent algorithms with our estimator can help base learning algorithms perform much better, especially on the OF1 and CF1 metrics.

Table 10: Comparison for classification performance with different loss correction ways on Pascal-VOC2007 dataset. The best peformances are in **bold**.

| | Noise rates $(\rho_-, \rho_+)$ | (0,0.2) | (0,0.6) | (0.2,0) | (0.6,0) | (0.1,0.1) | (0.2,0.2) | (0.017,0.2) | (0.034,0.4) |
|---|---|---|---|---|---|---|---|---|---|
| mAP | Reweight-Ours | 84.43±0.46 | 78.72±0.41 | 84.08±0.24 | **74.46±0.56** | 84.03±0.29 | 80.44±0.52 | 84.09±0.62 | 80.97±1.03 |
| | Backward-Ours | 83.41±0.18 | 67.22±2.97 | 81.13±0.45 | 63.82±1.36 | 79.00±0.88 | 70.44±1.80 | 81.27±0.37 | 69.22±1.96 |
| | Forward-Ours | 84.96±0.28 | **80.23±0.25** | 83.41±0.72 | 71.45±3.65 | 83.19±0.24 | 76.77±2.14 | 84.31±0.51 | 80.46±1.02 |
| | Revision-Ours | **84.99±0.33** | 79.26±0.51 | **84.37±0.24** | 74.44±1.16 | **84.49±0.36** | **80.61±0.73** | **84.98±0.11** | **82.14±0.10** |
| OF1 | Reweight-Ours | 78.62±0.58 | 65.68±1.67 | 80.85±0.25 | 67.43±4.65 | 79.64±0.29 | 75.52±0.86 | 79.25±0.52 | 74.35±1.65 |
| | Backward-Ours | 74.98±0.56 | 16.35±5.06 | 78.55±0.33 | 14.57±0.29 | 74.35±1.12 | 63.60±4.39 | 71.68±0.78 | 36.38±4.13 |
| | Forward-Ours | **80.09±0.45** | **70.38±0.29** | 80.44±0.50 | 59.35±10.36 | 79.77±0.35 | 73.92±2.33 | **79.92±0.62** | 75.18±1.24 |
| | Revision-Ours | 79.07±0.94 | 63.97±3.14 | **81.20±0.32** | **68.74±4.18** | **80.22±0.40** | **75.68±1.16** | 79.83±0.33 | **75.23±0.56** |
| CF1 | Reweight-Ours | 76.86±0.48 | 61.29±1.94 | 77.89±0.42 | **66.79±2.50** | 78.04±0.40 | **74.08±0.79** | 77.28±0.48 | **72.18±0.74** |
| | Backward-Ours | 70.64±0.59 | 15.01±4.66 | 75.48±0.84 | 14.55±0.26 | 66.26±2.15 | 46.19±8.35 | 65.56±1.59 | 23.78±2.95 |
| | Forward-Ours | **77.85±0.56** | **64.64±1.69** | 77.14±0.35 | 62.26±3.06 | 77.72±0.25 | 70.45±4.38 | 77.74±0.80 | 71.76±1.45 |
| | Revision-Ours | 76.82±0.90 | 58.56±3.72 | **78.26±0.51** | 66.89±1.46 | **78.55±0.58** | 73.94±1.06 | **77.86±0.33** | 72.12±1.16 |

Table 11: Comparison for classification performance with different base learning algorithms on Pascal-VOC2007 dataset. The best peformances are in **bold**.

| | Noise rates $(\rho_-, \rho_+)$ | (0,0.2) | (0,0.6) | (0.2,0) | (0.6,0) | (0.1,0.1) | (0.2,0.2) | (0.017,0.2) | (0.034,0.4) |
|---|---|---|---|---|---|---|---|---|---|
| mAP | Standard | 84.25±1.07 | 77.16±0.94 | 82.70±0.54 | 68.65±1.57 | 83.07±0.45 | 78.87±0.52 | 83.92±0.59 | 80.97±0.42 |
| | AGCN | 83.24±0.67 | 75.50±0.56 | 81.09±0.51 | 66.47±1.29 | 81.09±0.48 | 73.79±0.76 | 82.21±0.42 | 76.55±1.11 |
| | CSRA | 85.11±0.51 | 79.47±1.22 | 82.93±0.65 | 67.36±2.25 | 83.69±0.69 | 78.10±0.53 | 84.94±0.36 | 81.51±0.14 |
| | Reweight-Ours | 84.43±0.46 | 78.72±0.41 | 84.08±0.24 | 74.46±0.56 | 84.03±0.29 | 80.44±0.52 | 84.09±0.62 | 80.97±1.03 |
| | AGCN-R-Ours | 85.07±0.56 | **80.35±0.69** | 83.41±0.68 | 66.07±4.32 | 83.79±0.41 | 78.18±2.53 | 84.38±0.30 | 80.76±0.91 |
| | CSRA-R-Ours | **85.83±0.53** | 77.64±3.21 | **84.66±0.60** | **75.41±1.77** | **84.68±0.44** | **81.01±0.57** | **85.74±0.29** | **82.64±0.35** |
| OF1 | Standard | 75.24±1.40 | 32.02±5.49 | 78.85±0.43 | 15.08±0.25 | 79.24±0.43 | 75.85±0.84 | 75.98±1.04 | 59.67±1.65 |
| | AGCN | 74.92±1.02 | 30.97±3.78 | 75.45±2.06 | 16.85±0.56 | 78.69±0.31 | 72.64±0.51 | 75.16±0.58 | 56.56±1.64 |
| | CSRA | 76.94±1.03 | 33.65±2.73 | 77.71±1.23 | 15.94±0.32 | 80.36±0.53 | 76.92±0.34 | 77.91±0.63 | 62.19±1.97 |
| | Reweight-Ours | 78.62±0.58 | 65.68±1.67 | 80.85±0.25 | **67.43±4.65** | 79.64±0.29 | 75.52±0.86 | 79.25±0.52 | 74.35±1.65 |
| | AGCN-R-Ours | 80.28±0.41 | **71.36±2.52** | 79.18±1.38 | 46.77±3.34 | 79.06±0.92 | 73.14±2.63 | 79.49±0.76 | 75.78±2.11 |
| | CSRA-R-Ours | **80.61±0.80** | 61.48±3.61 | **81.22±0.45** | 57.48±9.33 | **80.43±0.44** | **76.04±0.80** | **81.22±0.34** | **77.21±0.45** |
| CF1 | Standard | 72.53±1.11 | 30.64±3.90 | 76.83±0.65 | 14.97±0.24 | 75.86±1.23 | 70.68±1.76 | 73.11±0.54 | 52.07±2.34 |
| | AGCN | 73.45±1.04 | 33.41±1.65 | 72.65±1.97 | 16.67±0.55 | 76.20±0.51 | 69.09±0.49 | 72.81±1.02 | 55.09±3.28 |
| | CSRA | 74.10±0.56 | 33.44±3.65 | 75.28±1.32 | 15.71±0.23 | 77.52±0.94 | 73.44±0.62 | 74.98±0.48 | 58.60±2.24 |
| | Reweight-Ours | 76.86±0.48 | 61.29±1.94 | 77.89±0.42 | **66.79±2.50** | 78.04±0.40 | 74.08±0.79 | 77.28±0.48 | 72.18±0.74 |
| | AGCN-R-Ours | **78.74±0.95** | **68.58±3.35** | 77.62±0.65 | 51.45±1.57 | 78.02±0.61 | 71.84±3.18 | 78.09±0.24 | 74.32±1.45 |
| | CSRA-R-Ours | 78.65±0.75 | 59.28±3.71 | **78.32±0.73** | 62.59±6.25 | **78.98±0.45** | **75.28±0.37** | **79.52±0.39** | **74.38±0.98** |

Besides, AGCN-R-Ours and CSRA-R-Ours perform better than Reweight-Ours in some cases, which means that the consistent algorithms with our estimator can work well with more advanced network.

## N  Discussion about Relaxation of Instance-independent Assumption

In this work, we assume that label noise is class-dependent but instance-independent. While, in the real-world scenarios, label noise is instance-dependent. Actually, this instance-independent assumption can be roughly relaxed to the assumption that the label noise of one class label is dependent on the label correlations with a few classes, and independent on the label correlations with most classes, which means most label pairs $(i, j)$s meet $P\left(\bar{Y}^j \mid Y^j, Y^i\right) = P\left(\bar{Y}^j \mid Y^j\right), P\left(\bar{Y}^i \mid Y^j, Y^i\right) = P\left(\bar{Y}^i \mid Y^i\right)$. With such labels, the system of equations involving $T^j$ in Section 3.2 holds, and our approach also works well. This relaxed assumption can be nearly satisfied in many real-world scenes, because generally speaking, the multi-label label noises for class $j$ are usually dependent on confusing features for itself, and the majority of classes will not share the same confusing features with class $j$. This claim agrees with the discussion about significantly different label pairs in Appendix D, and some research works about real-world label noise [43, 41] also show that noisy labels usually flips to some similar class labels in the real-world scene. For example, In CIFAR-100N, which is a re-annotated version of the CIFAR-100 with real-world human annotations, most classes are more likely to be mislabeled into less than four fine classes [43]. In ANIMAL-10N, the label noise mainly happens between five pairs of confusing animals [41]. Besides, the experiments in Appendix O also verify the effectiveness of our approach in two typical instance-dependent multi-label noise cases.

## O  Experiments on Instance-dependent Label Noise

We perform the experiments with two types of instance-dependent label noise: pair-wise label noise [13] and PMD label noise [59] on Pascal-VOC2007 and MS-COCO datasets to illustrate the

applicability in realistic scenarios. For pair-wise label noise, one class label is mistaken as another class label with a certain probability. For PMD label noise, data near the decision boundary are harder to distinguish and more likely to be mislabeled. Both of them are much realistic. The estimation error between the estimated transition matrices and $P\left(\bar{Y}^j \mid Y^j\right)$ can be seen in Tab. 12 and 13, and comparison for classification performance can be seen in Tab. 14 and 15. Note that as class labels in both datasets are unbalanced, in order to prevent class change, we only test with pair-wise label noise less than 20%, and do not flip the labels into classes with few positive examples. The results show the proposed estimator can achieve the smaller estimation errors on such instance-dependent cases, leading to better classification performance.

Table 12: Comparison for estimation error between the estimated transition matrices and $P\left(\bar{Y}^j \mid Y^j\right)$ on Pascal-VOC2007 dataset with instance-dependent label noise.

| Noise type | Pair-wise 10% | Pair-wise 15% | Pair-wise 20% | PMD-Type-I | PMD-Type-II | PMD-Type-III |
|---|---|---|---|---|---|---|
| T-estimator max | 1.99±0.02 | 2.97±0.01 | 3.95±0.01 | 8.70±0.13 | 5.66±0.02 | 6.13±0.02 |
| T-estimator 97% | 5.22±0.02 | 5.53±0.10 | 5.08±0.09 | 3.45±0.13 | 4.01±0.16 | 4.02±0.03 |
| Dual T-estimator max | **1.06±0.03** | **1.36±0.09** | 1.62±0.06 | 4.31±0.42 | 3.52±0.07 | 4.33±0.04 |
| Dual T-estimator 97% | 14.49±0.02 | 14.13±0.05 | 13.47±0.04 | 9.35±0.01 | 10.78±0.01 | 10.72±0.01 |
| Our estimator | 1.82±0.05 | 2.19±0.03 | **1.55±0.04** | **1.29±0.07** | **1.71±0.16** | **1.96±0.20** |

Table 13: Comparison for estimation error between the estimated transition matrices and $P\left(\bar{Y}^j \mid Y^j\right)$ on MS-COCO dataset with instance-dependent label noise.

| Noise type | Pair-wise 10% | Pair-wise 15% | Pair-wise 20% | PMD-Type-I | PMD-Type-II | PMD-Type-III |
|---|---|---|---|---|---|---|
| T-estimator max | **8.65±0.05** | 12.11±0.12 | 15.41±0.18 | 42.38±0.15 | 32.36±0.12 | 36.44±2.14 |
| T-estimator 97% | 53.35±0.07 | 49.61±0.12 | 44.87±0.08 | 20.32±0.02 | 33.32±0.06 | 30.83±2.03 |
| Dual T-estimator max | 9.15±0.21 | **7.58±0.04** | **6.76±0.19** | **14.06±0.13** | 18.57±0.12 | 21.53±1.26 |
| Dual T-estimator 97% | 69.31±0.03 | 64.91±0.02 | 60.38±0.02 | 31.45±0.04 | 44.75±0.01 | 40.60±2.35 |
| Our estimator | 8.81±0.05 | 9.17±0.41 | 8.78±0.16 | 15.67±0.21 | **16.26±0.07** | **19.70±2.44** |

Table 14: Comparison for classification performance on Pascal-VOC2007 dataset with instance-dependent label noise. The best peformances are in **bold**.

| | Noise type | Pair-wise 10% | Pair-wise 15% | Pair-wise 20% | PMD-Type-I | PMD-Type-II | PMD-Type-III |
|---|---|---|---|---|---|---|---|
| mAP | Standard | **85.32±0.09** | 83.19±0.05 | 82.06±0.34 | 78.03±0.42 | **82.98±0.36** | **82.09±0.69** |
| | Reweight-T max | 84.67±0.24 | 82.75±0.20 | 81.54±0.06 | 77.74±0.69 | 82.60±0.30 | 81.92±0.59 |
| | Reweight-T 97% | 84.54±0.12 | 83.08±0.34 | 81.17±0.82 | 78.04±0.73 | 82.32±0.59 | 81.88±0.54 |
| | Reweight-DualT max | 84.92±0.15 | **83.42±0.14** | 82.29±0.05 | 78.20±0.77 | 82.79±0.27 | 81.71±1.04 |
| | Reweight-DualT 97% | 83.87±0.24 | 77.97±0.17 | 75.50±0.11 | 75.46±0.62 | 80.02±0.22 | 80.39±0.28 |
| | Reweight-Ours | 84.75±0.08 | 83.34±0.11 | 82.50±0.03 | **78.52±0.65** | 82.50±0.29 | 81.85±0.41 |
| OF1 | Standard | 80.71±0.17 | 78.69±0.13 | 77.12±0.34 | 57.55±2.64 | 77.07±0.10 | 75.93±0.38 |
| | Reweight-T max | 80.48±0.34 | 77.75±0.09 | 76.68±0.16 | 63.58±3.06 | 77.42±0.57 | 76.33±0.92 |
| | Reweight-T 97% | 78.79±0.10 | 76.00±0.32 | 73.01±0.84 | 72.00±1.04 | 77.07±0.47 | 76.63±0.32 |
| | Reweight-DualT max | **80.96±0.36** | 78.89±0.22 | 78.87±0.19 | 69.30±3.85 | 77.81±0.57 | 76.77±0.78 |
| | Reweight-DualT 97% | 70.59±0.19 | 59.64±0.37 | 57.40±0.18 | 60.17±1.18 | 67.20±0.32 | 70.03±2.10 |
| | Reweight-Ours | 80.87±0.24 | **79.54±0.02** | **79.70±0.04** | **73.16±2.55** | **78.02±0.37** | **77.26±0.63** |
| CF1 | Standard | 77.88±0.13 | 75.34±0.26 | 73.10±0.82 | 51.95±2.43 | 74.07±0.60 | 72.65±0.79 |
| | Reweight-T max | 77.73±0.39 | 75.43±0.17 | 74.36±0.13 | 58.70±0.95 | 74.34±0.57 | 73.15±0.77 |
| | Reweight-T 97% | 77.67±0.29 | 75.95±0.33 | 73.92±1.02 | **72.34±0.41** | 75.83±0.21 | **75.27±0.34** |
| | Reweight-DualT max | 78.49±0.39 | 76.64±0.18 | 75.57±0.88 | 65.35±2.36 | 74.90±0.65 | 73.86±0.67 |
| | Reweight-DualT 97% | 73.90±0.14 | 65.79±0.10 | 60.23±0.18 | 62.28±1.36 | 71.22±1.00 | 71.50±1.02 |
| | Reweight-Ours | **78.84±0.24** | **77.56±0.06** | **77.49±0.09** | 72.17±1.64 | **76.22±0.45** | 74.98±0.62 |

# P   The Wilcoxon Signed-Ranks Test for Reweight-Ours against Baselines

Wilcoxon signed-ranks test [10] is employed to show whether Reweight-Ours has a significant performance than other comparing approaches. Note that the performance of these methods used for the Wilcoxon signed-ranks test is from both class-dependent label noise cases and instance-dependent label-noise cases. As shown in Tab. 16, Reweight-Ours outperforms other baselines on both OF1 and CF1 metrics at 0.1 significance level.

Table 15: Comparison for classification performance on MS-COCO dataset with instance-dependent label noise. The best peformances are in **bold**.

| | Noise type | Pair-wise 10% | Pair-wise 15% | Pair-wise 20% | PMD-Type-I | PMD-Type-II | PMD-Type-III |
|---|---|---|---|---|---|---|---|
| mAP | Standard | 70.48±0.12 | 68.85±0.10 | 67.65±0.20 | 61.06±0.03 | 65.69±1.27 | **64.46±0.23** |
| | Reweight-T max | 70.31±0.10 | 69.04±0.11 | 67.82±0.32 | 60.72±0.23 | **65.86±1.25** | 64.36±0.16 |
| | Reweight-T 97% | 68.54±0.07 | 66.53±0.46 | 65.02±0.11 | 59.40±0.01 | 63.96±0.91 | 62.55±0.22 |
| | Reweight-DualT max | 68.36±0.17 | 66.32±0.17 | 66.60±0.30 | 60.82±0.30 | 63.71±1.63 | 62.47±0.28 |
| | Reweight-DualT 97% | 64.34±0.26 | 61.80±0.55 | 59.75±0.04 | 53.51±0.09 | 59.82±0.06 | 58.55±0.28 |
| | Reweight-Ours | **70.84±0.09** | **69.37±0.05** | **68.31±0.01** | **61.76±0.08** | 65.33±0.78 | 64.14±0.07 |
| OF1 | Standard | 70.22±0.26 | 68.59±0.51 | 68.26±0.58 | 47.37±2.69 | 63.01±2.17 | 58.95±0.59 |
| | Reweight-T max | 70.41±0.02 | 69.44±0.10 | 68.32±0.52 | 55.96±1.89 | 63.82±2.04 | 59.93±0.13 |
| | Reweight-T 97% | 55.68±0.18 | 54.97±0.21 | 54.95±0.03 | 52.42±0.15 | 53.37±0.91 | 53.02±0.59 |
| | Reweight-DualT max | 67.02±0.23 | 66.06±0.34 | 66.47±0.62 | 61.63±0.33 | 61.24±3.67 | 58.92±0.77 |
| | Reweight-DualT 97% | 35.64±0.14 | 34.40±0.73 | 33.70±0.11 | 31.94±0.24 | 34.28±0.54 | 34.30±0.37 |
| | Reweight-Ours | **71.01±0.08** | **69.52±0.35** | **69.40±0.04** | **63.15±0.81** | **65.38±0.34** | **63.02±0.35** |
| CF1 | Standard | 64.81±0.37 | 62.40±0.48 | 61.75±0.65 | 41.09±1.64 | 56.40±2.58 | 52.12±0.47 |
| | Reweight-T max | 65.44±0.12 | 63.82±0.00 | 62.20±0.77 | 48.02±1.66 | 58.15±2.00 | 53.33±0.13 |
| | Reweight-T 97% | 54.26±0.05 | 52.54±0.04 | 52.08±0.10 | 50.12±0.33 | 52.58±0.69 | 51.93±0.04 |
| | Reweight-DualT max | 66.12±0.22 | 65.14±0.30 | 65.53±0.34 | 60.21±0.01 | 60.41±1.85 | 58.17±0.29 |
| | Reweight-DualT 97% | 38.44±0.03 | 37.24±0.41 | 36.84±0.01 | 32.74±0.54 | 38.50±0.40 | 37.84±0.32 |
| | Reweight-Ours | **68.58±0.03** | **67.81±0.07** | **66.90±0.14** | **60.82±0.11** | **62.34±1.82** | **60.05±0.22** |

Table 16: Summary of the Wilcoxon signed-ranks test for Reweight-Ours against other comparing approaches at 0.1 significance level. The p-values are shown in the brackets.

| Reweight-Ours against | Standard | GCE | CDR | AGCN | CSRA | WSIC | Reweight-T max | Reweight-T 97% | Reweight-DualT max | Reweight-DualT 97% |
|---|---|---|---|---|---|---|---|---|---|---|
| mAP | tie [0.29] | tie [0.26] | tie [0.32] | tie [0.18] | tie [0.45] | tie [0.26] | tie [0.32] | tie [0.23] | tie [0.35] | **win** [0.02] |
| OF1 | **win** [0.00] | **win** [0.05] | **win** [0.04] | **win** [0.02] | **win** [0.05] | **win** [0.02] | **win** [0.01] | **win** [0.08] | **win** [0.09] | **win** [0.00] |
| CF1 | **win** [0.02] | **win** [0.01] | **win** [0.01] | **win** [0.01] | **win** [0.02] | **win** [0.00] | **win** [0.03] | **win** [0.02] | **win** [0.09] | **win** [0.00] |

## Q  Broader Impact

In this era of deep learning, datasets are becoming bigger and bigger. However, it is much expensive to obtain large-scale dataset with high-quality annotated labels. Alternatively, people can collect labels based on non-expert annotators or automated labeling methods, e.g. web search and user tags, especially for non-profit research institutions and startups. However, these cheap annotation methods are likely to introduce noisy labels to the collected datasets, which may pose serious threats to the supervised learning algorithm designed for clean data. Label-noise learning, therefore, becomes a more and more important topic recently.

Existing label-noise learning methods typically focus on the single-label case by assuming that only one label is corrupted. In real applications, an example is usually associated with multiple class labels, which could be corrupted simultaneously with their respective different probabilities. The proposed method is to estimate the transition matrix using label correlations in noisy multi-label datasets. The transition matrix is essential to building the statistically consistent label-noise learning algorithms. We have shown that our method usually leads to a better estimation compared to the current estimator and can improve the classification performance of statistically consistent label-noise learning applications, which should has a positive impact on science, society, and economy. Hence, our approach can reduce the need for accurate labeling, and potentially, it may have a negative impact on the salary of label workers.