# OpenReview forum: "Estimating Noise Transition Matrix with Label Correlations for Noisy Multi-Label Learning"
_NeurIPS.cc/2022/Conference — NeurIPS 2022 Accept_

### Official Review · Reviewer_n5cR · 2022-07-10

**Rating:** 6
**Confidence:** 1
**Soundness:** 3 good
**Presentation:** 2 fair
**Contribution:** 3 good

**Summary:**

This paper discusses the estimation problem of the transition matrices in the noisy multi-label setting. They study the identifiability problem of the class-dependent transition matrix in noisy multi-label learning and propose a new estimator by exploiting label correlations without both anchor points and accurate fitting of noisy class posterior inspired by the identifiability results. Experimentally, the effectiveness of the proposed method is also illustrated.

**Questions:**

1. Can the authors clarify the originality of the proof of Theorem 1-4?
2. Some dots should be replaced by $\cd$, e.g. the dots in line 190 and line 204.

**Limitations:**

They address it adequately in Appendix.

**Strengths And Weaknesses:**

**Strenghths:**
1. They utilize the mismatch of label correlations to identify the transition matrix without both anchor points and accurate fitting of noisy class posterior in noisy multi-label learning.
2. The method is effective to the issue with both theoretical analyses and empirical results support this method.
3. Clear writing and structure.

**Weaknesses:**
1. Some formulas like in line 192 are recommended to be written in more simplified way.
2. The experimental results are recommended to be presented in different forms but not just tables.

---

> ### Author Response · Authors · 2022-08-02
> **Response to Reviewer n5cR**
>
> Thank you very much for your kind comments on this paper. We address your concerns as follows.
>
> > **Q1:** Some formulas like in line 192 are recommended to be written in more simplified way.
>
> **A1:** Thank you for your suggestion. The noisy multi-label case scenario is different from the noisy multi-class case, and each class label $Y^j$ has corresponding noise labels $\bar{Y}^j$ and transition matrices $T^j$, which may make our formulas less clear.
>
> > **Q2:** The experimental results are recommended to be presented in different forms but not just tables.
>
> **A2:** Following your kind suggestion, we have added some charts to show more experimental results in the appendices, and we have updated the revised version.
>
> > **Q3:** Can the authors clarify the originality of the proof of Theorem 1-4?
>
> **A3:** The proof of Theorem 1 and 4 is established directly based on the basic properties of probability and matrix, where we prove them by transforming the probability problem into the decomposition problem of a matrix. The proof of Theorem 2 and 3 is based on some Kruskal’s identiability results from [1-4].   Liu et al.[3] first built identifiability of the noise transition matrix based on the Kruskal’s identifiability results in the noisy multi-class learning. Inspired by them, we use the same Kruskal’s identiability results to prove that If $\bar{Y}^i$ and $\bar{Y}^k$are independent given ${Y}^j$, $T^j$ is identifiable, and if not, it is unidentifiable.
>
> > **Q4:** Some dots should be replaced by \cd, e.g. the dots in line 190 and line 204.
>
> **A4:**  Thanks very much for your correction.  We have fixed them in the revised version, and we highly appreciate your careful comments.
>
>
> [1] Three-way arrays: rank and uniqueness of trilinear decompositions, with application to arithmetic complexity and statistics. Linear algebra and its applications, 1977.
>
> [2] On the uniqueness of multilinear decomposition of  n-way arrays. Journal of Chemometrics: A Journal of the Chemometrics Society. 2000.
>
> [3] Identifiability of label noise transition matrix. arXiv, 2022.
>
> [4] The analysis of three-way arrays by constrained PARAFAC methods.  DSWO Press, Leiden University, 1993.

---

> > ### Comment · Reviewer_n5cR · 2022-08-09
> > **Response to Authors**
> >
> > Thank you very much for your feedback. They address my concerns.

---

> > > ### Author Response · Authors · 2022-08-09
> > > **Comment to Reviewer n5cR**
> > >
> > > Thanks again for your kind comments. We will carefully consider your suggestions to further revise our paper.

---

> ### Author Response · Authors · 2022-08-06
> **Comment to Reviewer n5cR**
>
> Thanks a lot for your efforts in reviewing this paper. We tried our best to address your mentioned concerns. Are there unclear explanations? We could further clarify them.

---

### Official Review · Reviewer_KMFh · 2022-07-11

**Rating:** 4
**Confidence:** 3
**Soundness:** 2 fair
**Presentation:** 2 fair
**Contribution:** 2 fair

**Summary:**


This paper presents a way of estimating the noise transition matrix for noisy multi-label learning, which makes use of label correlations through a bilinear decomposition based on frequency counting. It first theoretically studies the identifiable problem of estimating the noise transition matrix under the multilevel setting,  which then leads to the development of the bilinear decomposition-based method for estimating the noise transition matrix.  Experiments results derived from several image datasets show that the proposed method can better estimate the transition matrix.

**Questions:**

1. Label correlation plays an important role in the proposed estimation. Even though assumption 2 is validated in the appendix. Could the authors empirically show this role with some examples from the experiments? For example, labels can be strongly correlated and or weakly correlated. Is it possible to study the association between the strength of label correlation and the model performance?
2. The server imbalance issue is indeed a challenging problem in multi-label learning, which I agree with the authors. I am wondering if the proposed method can deal with this issue? If so, how?
3. Other common performance metrics used in multiple-label learning are the ranking-based metrics, like P@K, R@K, NDCG@k, etc. Could the authors consider one of the metrics, say NDCG@K?
4. How will the threshold $\tau$ affect the model performance?
5. Given the mixed performance of the proposed method,  Is there any consistent takeaway message here? I.e., In which circumstance the proposed method can be applied and perform better?
6. What is $n_a$ in computing $\rho{-}$ in line 222?

**Limitations:**

The authors have addressed the limitations and border impact in the appendix.

**Strengths And Weaknesses:**

Strengths

Label-nosing learning is an important and challenging problem not just in multi-class classification and also in the multi-label setting.  This paper presents a bilinear decomposition-based approach based a simple frequency counting with a clearly described algorithm.

The proposed estimation method is well motivated with a set of theorems that study the identifiable problem in the multi-label learning setting, which seems to be thorough with proper proof in the Appendix. The proof is also easy to follow.

Overall, the paper is written well, and most of the notations were clear and well-organized to guide the readers through the text.


Weaknesses

My major concern is around the performance of the proposed estimation method.  In terms of estimation of the transition matrices, it outperforms the competitors in most of the settings, which is good. However, when coming to the classification performance, the proposed method is not a clear winner, particularly compared with those without considering noise labels across the datasets. For instance, for varying levels of noise, it is hard to conclude the behaviour of the proposed method.

Ablation studies showing how the proposed model benefit from label correlation and deal with imbalance issue are missing. For example, there is a running parameter $\tau$ used in Stage 1 to select the sample set $\mathcal{D}^j_t$, which I believe will have an impact on the performance of the proposed method.

---

> ### Author Response · Authors · 2022-08-02
> **Response to Reviewer KMFh Part (1)**
>
> Thank you very much for your comments on this paper. We address your concerns as follows.
>
> > **Q1:** The concern around the mixed performance of the proposed method on estimation of the transition matrices and classification.  In which circumstance the proposed method can be applied and perform better?
>
> **A1:**  (1) Theoretically, a statistically consistent algorithm with accurate transition matrices can guarantee to vanish the differences between the classifiers learned from noisy data and the optimal ones from clean data by increasing the size of noisy examples [1,2]. When the size of noisy data is infinite, these consistent algorithms can guarantee to obtain the optimal classifier. While, in the real-world datasets, the size of noisy data is finite, and therefore,  a statistically consistent algorithm with more accurate transition matrices will obtain a classifier close to the optimal one with a higher probability. That is to say, when transition matrices become more accurate, the learned classifier becomes more likely to be better, but does not always become better. It is the main reason for the mixed performance of the proposed method on the estimation of the transition matrices and classification. Besides, the probability that a statistically consistent algorithm with accurate transition matrices obtains a classifier close to the optimal one, will become higher by increasing the size of noisy data. Hence, if our method can achieve a more accurate estimation, a statistically consistent algorithm with our estimator will be more likely to be better in various cases,  and its advantage will become more significant when the size of noisy data is large.
>
> (2) Our experimental results accord with the above discussion. As seen in Section 4.2, on all datasets, among those consistent methods, Rewight algorithm with our estimator (named "Reweight-Ours") obtains the most best or second-best results on the three metrics, which is due to the smaller error of our estimation. On MS-COCO dataset, which is much larger than VOC2007 and VOC2012 datasets, Rewight algorithm with our estimator achieves a more significant and stable advantage among the consistent methods.
>
> (3) As shown in Section 4, when compared with those methods without considering noise labels across the datasets,  our method can perform much better on the OF1 and CF1 metrics, which shows the model learned by our method can approximate the true class posterior well. Besides, our method can also help some methods without considering noise labels, e.g. CSRA and AGCN, perform much better in noisy multi-label learning,  which has been shown in Appendix L.
>
> (4) Noise transition matrices can not only be used in statistically consistent algorithms, but also can be used to estimate noise rate for noise detection methods [3,4].  We believe an accurate estimation method for transition matrices in multi-label cases can promote the research of noise detection methods for noisy multi-label learning.
>
>
>
> > **Q2:** Ablation studies showing how the proposed model benefits from label correlation
>
> **A2:** Thank you for your suggestion. According to the results in Appendix B, we divide all labels (except class label 50) on MS-COCO dataset into two categories: one is very likely to have a correlation with class label 50  (the label belonging to this category is termed as "label A")  and the other is less likely to have a correlation with class label 50  (the label belonging to this category is termed as "label B") . Table 3-1 shows the mean estimation error of transition matrices for class label 50 by using the correlations of label 50 and label A (or B) in various cases.  The results represent the estimation error will be smaller using the correlations with label A,  which means stronger label correlations can lead to better estimation in our approach.
>
> | Estimation Error           |       (0,0.2)       |       (0.2,0)       |     （0.1,0.1)      |    （0.017,0.2)     |  Average  |
> | -------------------------- | :-----------------: | :-----------------: | :-----------------: | :-----------------: | :-------: |
> | using label 50 and label A | **0.025$\pm$0.024** | **0.090$\pm$0.049** | **0.106$\pm$0.051** | **0.099$\pm$0.040** | **0.080** |
> | using label 50 and label B |   0.057$\pm$0.062   |   0.139$\pm$0.111   |   0.198$\pm$0.123   |   0.172$\pm$0.104   |   0.269   |
>
> Table 3-1: Mean estimation error of  transition matrices for class label 50 on MS-COCO dataset using the correlations with different labels.
>
> [1] Are Anchor Points Really Indispensable in Label-Noise Learning? NeurIPS 2019.
>
> [2] Classification with noisy labels by importance reweighting. TPAMI 2016.
>
> [3] How does disagreement help generalization against label corruption? ICML 2019.
>
> [4] Detecting Corrupted Labels Without Training a Model to Predict. ICML 2022.

---

> > ### Author Response · Authors · 2022-08-02
> > **Response to Reviewer KMFh Part (2)**
> >
> > > **Q3:** Lack of ablation study about the sample selection threshold $\tau$.
> >
> > **A3:** (1)The sample selection we adopted is to estimate this clean probability of samples by modeling sample loss values with a GMM model [5,6] using the Expectation-Maximization algorithm.  If the clean sample can be distinguished according to loss values, and its estimated probability is accurate, the best threshold will be about 0.5. Hence, $\tau=0.5$ is a typical value in related works [5,6], and we follow this practice in our experiments.
> >
> > (2) Using the classification performance on noisy validation set as the criterion for model selection is a typical and empirically useful practice [7-10] in label-noise learning, even in the cases with instance-dependent label noise [9,10]. In this paper, we use mAP score on the noisy validation set as the criterion for model selection. Table 2-1 shows the ablation study about $\tau$, which represents $\tau=0.5$ is a good choice  both according to mAP scores on the noisy validation set and according to mAP scores on the clean test dataset.
> >
> > | mAP (validation/test) |                (0,0.2)                | （0.2,0）                         | （0.1,0.1)                            | （0.017,0.2)                      | Average             |
> > | --------------------------------------------------- | :-----------------------------------: | --------------------------------- | ------------------------------------- | --------------------------------- | ------------------- |
> > | $\tau=0.4$                                          |     71.49$\pm$1.01/84.09$\pm$0.62     | 48.37$\pm$0.55/83.83$\pm$0.23     | 44.02$\pm$0.82/83.69$\pm$0.68         | **56.45$\pm$1.97**/84.30$\pm$0.32 | 55.08/83.98         |
> > | $\tau=0.5$                                          | **72.50$\pm$0.68**/**84.77$\pm$0.40** | 48.06$\pm$0.58/**83.92$\pm$0.16** | **44.30$\pm$1.54**/**84.01$\pm$0.46** | 56.16$\pm$1.90/84.36$\pm$0.59     | **55.26**/**84.27** |
> > | $\tau=0.6$                                          |     71.44$\pm$1.00/84.18$\pm$0.65     | **48.38$\pm$0.57**/83.84$\pm$0.29 | 44.00$\pm$0.86/83.70$\pm$0.66         | 56.37$\pm$1.90/**84.37$\pm$0.29** | 55.05/84.01         |
> >
> > Table 3-2: Ablation study about the sample selection threshold $\tau$ on VOC2007 datasets.
> >
> >
> >
> >
> >
> > > **Q4:** Can the proposed method deal with the server imbalance issue?
> >
> > **A4:** For most of the estimation methods of the transition matrix, they need to accurately estimate the noisy posterior probability. The server positive-negative class imbalance, making it difficult for the networks to accurately estimate the noisy posterior probability.   Since our estimator utilizes label correlations to perform transition matrix estimation, which does not need to accurately estimate the noisy posterior probability,  it naturally avoids this problem in the transition matrix estimation.
> >
> >
> >
> > > **Q5:** Can authors report other common performance metrics used in multiple-label learning are the ranking-based metrics, like P@K, R@K, NDCG@k?
> >
> > **A5:**  Thank you for your advice. However, P@K, R@K, and NDCG@k are the metrics used for retrieval task, and our work is about multi-label classification learning.  The evaluation metrics we reported are widely in related works [11,12,13], including the mean average precision (mAP),  overall F1-measure (OF1) and per-class F1-measure (CF1) .  Actually, the mAP metric is a ranking-based metric [14,15], which only depends on the order of prediction confidence for labels.
> >
> >
> >
> > > **Q6:** What is $n_a$ in computing?
> >
> > **A6:** $n_a$ is the cardinality of original multi-label dataset, which refers to the average number of labels appearing in one instance, and we have defined it when introducing datasets in Section 4.
> >
> > [5] Unsupervised Label Noise Modeling and Loss Correction. ICML 2019.
> >
> > [6] DivideMix: Learning with Noisy Labels as Semi-supervised Learning. ICLR 2020.
> >
> > [7] Robustness of Accuracy Metric and its Inspirations in Learning with Noisy Labels. AAAI 2021.
> >
> > [8] Are Anchor Points Really Indispensable in Label-Noise Learning? NeurIPS 2019.
> >
> > [9] Parts-dependent Label Noise: Towards Instance-dependent Label Noise. NeurIPS 2020.
> >
> > [10] A Second-Order Approach to Learning With Instance-Dependent Label Noise, CVPR 2021.
> >
> > [11] Multi-label image recognition with graph convolutional networks. CVPR 2019.
> >
> > [12] Attention-driven dynamic graph convolutional network for multi-label image recognition. ECCV 2020.
> >
> > [13] Asymmetric loss for multi-label classification. ICCV 2021.
> >
> > [14] A Review on Multi-Label Learning Algorithms. TKDE 2014.
> >
> > [15] The PASCAL Visual Object Classes Challenge 2007 (VOC2007).

---

> ### Author Response · Authors · 2022-08-06
> **Comment to Reviewer KMFh**
>
> Thanks a lot for your efforts in reviewing this paper. We tried our best to address your mentioned concerns. Are there unclear explanations? We could further clarify them.

---

> ### Author Response · Authors · 2022-08-07
> **Comment to Reviewer KMFh**
>
> Thank you for reviewing this paper. There are less than three days left until the end of the author-reviewer discussion. We hope you can give some feedback on our response so that we can further explain it if something is unclear.

---

> ### Author Response · Authors · 2022-08-08
> **Comment to Reviewer KMFh**
>
> Following your good suggestions, we have added ablation studies about label correlation and the sample selection threshold, and updated them in the revised version. Is there something unclear in our response and revision? We would like to further explain it.

---

### Official Review · Reviewer_9BSQ · 2022-07-11

**Rating:** 7
**Confidence:** 5
**Soundness:** 3 good
**Presentation:** 2 fair
**Contribution:** 3 good

**Summary:**

The paper proposes a new estimator to estimate the transition matrix in noisy multi-label learning. The main idea of the estimator is to utilize the clean label co-occurrence as a constraint to help the estimation of the transition matrix. Specifically, authors derive a two-stage method to estimate then transition probabilities. At the first stage, the paper utilizes an existing technique to select clean labels for estimating the label occurrence. Although the estimated label occurrence is biased due to the selection bias, authors claims that the selection biased cannot lead to large estimation error. At the section stage, the paper obtains the label co-occurrence with frequency counting and use the mismatch of label correlation to estimate the transition matrix.

**Questions:**

How to determine the threshold \tau, especially without the clean validation set?

**Ethics Review Area:**

["I don’t know"]

**Strengths And Weaknesses:**

Strength

1. The motivation of the proposed method is reasonable. The information of label co-occurrence is useful for the transition matrix estimation.

2. The paper reports extensive experiments to validate the effectiveness of the proposed method.

3. The paper conducts comprehensive analyses for noisy multi-label learning.


Weakness

1. In section 3.2, authors claim that the label co-occurrence estimated by using selected clean labels is unbiased. This is based on the assumption that “given Y^j”, the features about class j is biased, while the features about another class i is unbiased. The assumption is unreasonable, since the features are biased with respect to class i and class j co-occur in an image.

2. Authors select a small number of clean labels for estimating the label co-occurrence. However, the estimated label co-occurrence may be imprecise due to the insufficient labels used for frequency counting, especially for positive labels.

3. There are many typos, such as page 4 line 168 Ys; Algorithm 1 line 4 Dt; page 2 line 48 “while “bird” and “sky” are always co-occurrence”.

4. The references are not cited properly. In experiments, the citations for multi-label with missing labels and partial multi-label are missed.

---

> ### Author Response · Authors · 2022-08-02
> **Response to Reviewer 9BSQ**
>
> Thank you very much for your comments on this paper. We address your concerns as follows.
>
> > **Q1:** The unbiased assumption is unreasonable.
>
> **A1:** We agree with you that this assumption will not strictly hold due to the complex label correlations, and this is the main error our approach leads to. But as we stated in Section 3.2, we claim that this bias for estimating $P(\bar{Y}^{i} \mid Y^{j})$ will not too large, since some simplest samples for each label can be easily selected without being affected by the presence or absence of other labels. And  In Section 4.1, our empirical results justify this by showing a little gap between the estimation error of our estimator with the biased sample selection and an unbiased one.
>
>
>
> > **Q2:** The estimated label co-occurrence may be imprecise due to the insufficient labels used for frequency counting
>
> **A2:**  We found this phenomenon in the experiments, and reported it as the limitations in Appendix G. Although this estimation error of  frequency counting will converge to zero exponentially fast [1], when the number of one label appearing is too small, e.g.  less than 50, the estimation of its transition matrix is still difficult to be accurate. We think it may be better to estimate the transition matrix using Dual-T estimator for this class label.
>
>
>
> > **Q3:** There are many typos.
>
> **A3：**Thank you for your concern.  We have fixed the typos and uploaded the revised version.
>
>
>
> > **Q4:** The references are not cited properly. In experiments, the citations for multi-label with missing labels and partial multi-label are missed.
>
> **A4:**  Thank you for your suggestion.  We have added some references about multi-label with missing labels [2,3] and partial multi-label learning [4,5] in the revised version.
>
>
>
> > **Q5:** How to determine the threshold $\tau$, especially without the clean validation set?
>
> **A5:** (1)The sample selection we adopted is to estimate this clean probability of samples by modeling sample loss values with a GMM model [6,7] using the Expectation-Maximization algorithm.  If the clean sample can be distinguished according to loss values, and its estimated probability is accurate, the best threshold will be about 0.5. Hence, $\tau=0.5$ is a typical value in related works [6,7], and we follow this practice in our experiments.
>
> (2) Using the classification performance on noisy validation set as the criterion for model selection is a typical and empirically useful practice [8-11] in label-noise learning, even in the cases with instance-dependent label noise [10,11]. In this paper, we use mAP score on noisy validation set as the criterion for model selection. Table 2-1 shows the ablation study about $\tau$, which represents $\tau=0.5$ is a good choice  both according to mAP scores on the noisy validation set and according to mAP scores on the clean test dataset.
>
> | mAP (validation/ test) |                (0,0.2)                | （0.2,0）                         | （0.1,0.1)                            | （0.017,0.2)                      | Average             |
> | --------------------------------------------------- | :-----------------------------------: | --------------------------------- | ------------------------------------- | --------------------------------- | ------------------- |
> | $\tau=0.4$                                          |     71.49$\pm$1.01/84.09$\pm$0.62     | 48.37$\pm$0.55/83.83$\pm$0.23     | 44.02$\pm$0.82/83.69$\pm$0.68         | **56.45$\pm$1.97**/84.30$\pm$0.32 | 55.08/83.98         |
> | $\tau=0.5$                                          | **72.50$\pm$0.68**/**84.77$\pm$0.40** | 48.06$\pm$0.58/**83.92$\pm$0.16** | **44.30$\pm$1.54**/**84.01$\pm$0.46** | 56.16$\pm$1.90/84.36$\pm$0.59     | **55.26**/**84.27** |
> | $\tau=0.6$                                          |     71.44$\pm$1.00/84.18$\pm$0.65     | **48.38$\pm$0.57**/83.84$\pm$0.29 | 44.00$\pm$0.86/83.70$\pm$0.66         | 56.37$\pm$1.90/**84.37$\pm$0.29** | 55.05/84.01         |
>
> Table 2-1: Ablation study about the sample selection threshold $\tau$ on VOC2007 datasets.
>
>
>
>
>
> [1]  Concentration inequalities - A nonasymptotic theory of independence. Concentration Inequalities, 2013.
>
> [2] Multi-Label Learning with Missing Labels. ICPR 2014.
>
> [3] Multi-label Learning with Missing Labels Using Mixed Dependency Graphs. IJCV 2018.
>
> [4] Partial Multi-Label Learning. AAAI 2018.
>
> [5] Partial Multi-Label Learning With Noisy Label Identification. TPAMI 2022.
>
> [6] Unsupervised Label Noise Modeling and Loss Correction. ICML 2019.
>
> [7] DivideMix: Learning with Noisy Labels as Semi-supervised Learning. ICLR 2020.
>
> [8] Robustness of Accuracy Metric and its Inspirations in Learning with Noisy Labels. AAAI 2021.
>
> [9] Are Anchor Points Really Indispensable in Label-Noise Learning? NeurIPS 2019.
>
> [10] Parts-dependent Label Noise: Towards Instance-dependent Label Noise. NeurIPS 2020.
>
> [11] A Second-Order Approach to Learning With Instance-Dependent Label Noise, CVPR 2021.

---

> > ### Comment · Reviewer_9BSQ · 2022-08-08
> > **Responses to Authors**
> >
> > Thanks for authors' feedback. The responses deal with most of my concerns. I agree that this is generally an interesting work.
> >
> > Due to the complex label correlations, does the equality still hold in Eq.(1)?

---

> > > ### Author Response · Authors · 2022-08-08
> > > **Responses to Reviewer 9BSQ**
> > >
> > > Thank you very much for your kind comments.  As for your question, we think the sample selection for class label j is mainly biased on the easy-to-discriminative features for itself,  and due to the complex label correlations, Eq.(1) will not hold when the instances of class labels i and j share the major discriminative features. The sample bias is the main factor that contributes to the error for our estimator. Thanks again for your efforts in reviewing this paper.

---

> > > > ### Comment · Reviewer_9BSQ · 2022-08-09
> > > > **Response to Authors**
> > > >
> > > > Thanks for your great efforts.
> > > >
> > > > A kind reminder is that Eq.(1) should be further considered and revised, since the equality may not hold in most cases.
> > > >
> > > > I decide to increase my rating. It is an interesting work although there still exist some problems with the manuscript.

---

> > > > > ### Author Response · Authors · 2022-08-09
> > > > > **Response to Reviewer 9BSQ**
> > > > >
> > > > > Thanks a lot for your kind reminder. We will further carefully consider Eq.(1) and add more explanations in the revised version.

---

> ### Author Response · Authors · 2022-08-06
> **Comment to Reviewer 9BSQ**
>
> Thanks a lot for your efforts in reviewing this paper. We tried our best to address your mentioned concerns. Are there unclear explanations? We could further clarify them.

---

### Official Review · Reviewer_hyUM · 2022-07-12

**Rating:** 5
**Confidence:** 5
**Soundness:** 2 fair
**Presentation:** 2 fair
**Contribution:** 2 fair

**Summary:**

This paper deals with how to estimate the 'noise transition matrix' under a multi-label setup. Like a single-label setup, the authors attempted to make an idea without using 'anchor points' for matrix estimation. They leverage the 'sample selection' method (ie., GMM) for extra clean supervision and provide a detailed mathematical derivation for the matrix estimation. The experimental section includes VOC and MS-COCO datasets; they injected a synthetic label noise with two factors, controlling the flip ratio of '1'->'0' and '0'->'1'. Under the proposed synthetic noise, the method works well and outperforms other simple extensions of similar works (but developed under the single-label setup).


**Questions:**

Please see the weakness section above.

**Limitations:**

Please see the weakness section above. In addition, the paper writing should be improved, particularly the theoretical section. Too many results are omitted and borrowed from [16, 24, 25, 31].

**Strengths And Weaknesses:**

### Strengths
1. This paper is the first approach to estimating the noisy transition matrix for handling noisy labels with multi-labels.
2. Leveraging mismatched label correlation is useful.
3. The method shows higher performance than other simple baselines.

### Weaknesses
I agree that estimating the noise transition matrix helps make a statistically consistent classifier. However, I felt some major weaknesses of this paper, as below:
1. **Assumption of Instance-independent Noise.** As the authors said, many previous studies make the same assumption for matrix estimations. But, there have been numerous recent studies to overcome this in the same direction [1-3]. I agree that dealing with 'instance-dependent' label noise in a multi-label setup is very difficult to address. At least, the author should mention what is more challenging in this setup compared to the single-label case. And, why making this assumption is reasonable with a more specific reason. Just saying 'the vast majority of current algorithms mainly focus' is not enough as a research paper.
2. **Noise Injection Protocol.** According to the paper, the authors assume the class-dependent label noise and aim to estimate an accurate noise transition matrix under this assumption. However, the noise injection protocol in Line 217 looks like 'class-independent label noise. Specifically, the $\rho just means the probability of '0' (or '1') being flipped to '1' (or '0') without any consideration of class pairs like <i, j>; only class j associates with the flipping and no connection between classes. Therefore, this is not very realistic and not class-dependent label noise. Did I get it wrong?
3. **Unrealistic Evaluation.** Related to the second weakness, the author should have included at least one realistic real noisy data. There are benchmark data with noisy multi-label instances, called OpenImages database. The author can find other better datasets if possible. Without realistic evaluation, I am not convinced of the robust results in the paper since the injection protocol is very unrealistic.
4. **Class Imbalance Problem & Low mAP.** Unlike single-label classification, the biggest difference in multi-label setup is 'class imbalance'. The authors also mentioned that this problem is very severe in Lines 44 & 102. However, there is no detailed mention of how to resolve this issue with the proposed idea throughout the paper. Next, when I am looking at the results on MS-COCO, the mAP value is too low compared with other multi-label papers using the same ResNet-50 pre-trained models. As an example, [4] shows ResNet-50 with GAP head achieves around 80mAP on MS COCO dataset (refer to Figure 6). However, in this paper, the mAP was less than 70 with Standard at the easiest noise setup (0.0, 0.2); this is a 20% missing label scenario. There are 20% missing labels but I think 20% of missing labels do not make 10mAP drops. Could you report the performance of your method and others under (0, 0) setup? This is a very good reference to check whether your implementation is correct and whether your method is still comparable with a zero-noise setup.

**[1]** Approximating instance-dependent noise via instance-confidence embedding, arXiv 2021

**[2]** A Second-Order Approach to Learning With Instance-Dependent Label Noise, CVPR 2021

**[3]** An Information Fusion Approach to Learning with Instance-Dependent Label Noise, ICLR 2022

**[4]** ML-Decoder: Scalable and Versatile Classification Head, arXiv 2021

---

> ### Author Response · Authors · 2022-08-02
> **Response to Reviewer hyUM Part (1)**
>
> Thank you very much for your comments on this paper. We address your concerns as follows.
>
> > **Q1:** The reasons for the assumption of instance-independent noise.
>
> **A1:** (1)  In the single-label case, given only noisy examples, the instance-dependent transition matrix $T_{ik}(X=x)=P(\bar{Y}=k|Y=i,X=x)$ is non-identifiable without any additional assumption[1], which also holds in the multi-label scenario. For example, $P(\bar{Y}^j=k|X=x)=\sum_{i=0}^1 T^j_{ik}P(Y^j=i|X=x)$ and $P(\bar{Y}^j=k|X=x)=\sum_{i=0}^1 T^{\prime j}_{ik} P^{\prime}(Y^j=i|X=x)$ are both valid, when
>
> $T_{ik}^{\prime j}(X=x)=T^j_{ik}(X=x) P(Y^j=i | X=x) / P^{\prime}(\bar{Y}^j=i | X=x)$.
>
>
> (2)The existing estimation methods [2-4] for instance-dependent transition matrices are hard to be applied in the multi-label case since these methods need to accurately approximate the noisy posterior probability of certain instances, which is very difficult in multi-label learning due to the positive-negative class imbalance. Besides,  PTD [2], the most famous instance-dependent matrix estimation algorithm, needs much more anchor points in the multi-label case than in the single-label case, making it not applicable, because it needs to decompose instances into much more parts with multiple objects in one instance, and estimate transition matrices for each class label.
>
>
>
> (3) Class-dependent transition matrices can not only be used in statistically consistent algorithms but also can be used to estimate the overall noise rate for noise detection methods [13,14].  We believe an accurate estimation method for class-dependent transition matrices in multi-label cases can promote the research of noise detection methods for noisy multi-label learning.
>
>
>
> > **Q2:** The noise injection protocol is more like "class-independent" label noise, and it is not realistic.
>
> **A2:** (1) In the single-label case, the "class-independent"  label noise means each label is flipped independently with a constant probability $\rho$, and the "class-dependent"  label noise for class label $Y$ means the flip probabilities $\rho_y$ are the same for all training labels when $Y=y$ [4]. Correspondingly, in this paper, the "class-dependent" label noise for class label $Y_j$ means that the flip probabilities are dependent on $Y^j=0$ or $Y^j=1$. The definition of this noise model is consistent with "Class-Conditional Multi-Label Noise" in [7],  and this type of noise injection protocol is also adapted in previous works [7,8] about noisy multi-label learning.
>
>
> (2) As stated in the paper, the noise injection protocol we used can simulate various situations, including multi-label learning with missing labels [8], partial multi-label learning [9], and class-conditional multi-label noise [7].
>
>
> (3)  We think that although the label noise model is independent on label correlations, this class-dependent model and the proposed estimation method are very useful in most realistic multi-label cases, because, unlike the single-label recognition task, most of the class labels are hardly confused by each other in multi-label learning,  such as  "bird"  vs "car",  "person" vs "clock",  "bear" vs"apple",  and "blue" vs "football".  These labels have correlations, i.e. $P\left(Y^{i}=0 | Y^{j}=0\right) \neq P\left(Y^{i}=0 | Y^{j}=1\right)$, but their label noise will be nearly independent on their clean label correlations in the real scene, i.e. $P\left(\bar{Y}^{j}| Y^{j}, Y^{i}\right) = P\left(\bar{Y}^{j}| Y^{j} \right), P\left(\bar{Y}^{i}| Y^{j}, Y^{i}\right) = P\left(\bar{Y}^{i}| Y^{i} \right)$, and these labels massively exist in MS-COCO and OpenImages datasets. With such labels,  the equation in Line 192 still holds and our approach also works well.  Besides, in the implementation of our estimator, we perform the R times estimation for each transition matrix, and get the final estimation by Eq.(7).  If the label noise of two labels is not independent on their correlations, they maybe can not get a meaningful probability estimation and will be discarded, or our approach will not choose the solution as the final estimation by Eq.(7), since it does not similar to other solutions. To further verify this statement, we introduce pair-wise label noise [10], which mistakes a class label with another label with a certain probability, into the VOC2007 datasets.  Comparison for classification performance can be seen in Table 1-1. Note that as class labels in VOC2007 dataset are unbalanced, in order to prevent class change, we only test with pair-wise label noise less than 20%.
>
> [1] Are Anchor Points Really Indispensable in Label-Noise Learning? NeurIPS 2019.
>
> [2] Parts-dependent Label Noise: Towards Instance-dependent Label Noise. NeurIPS 2020.
>
> [3] A Second-Order Approach to Learning With Instance-Dependent Label Noise, CVPR 2021.
>
> [4] Learning with Bounded Instance- and Label-dependent Label Noise. ICML 2020.
>
> [5] Making deep neural networks robust to label noise: A loss correction approach. CVPR 2017.

---

> > ### Author Response · Authors · 2022-08-02
> > **Response to Reviewer hyUM Part (2)**
> >
> > | mAP(%)/ OF1(%) / CF1(%) | Pair-wise 10%                                    | Pair-wise 15%                                            | Pair-wise 20%                                            |
> > | ----------------------- | ------------------------------------------------ | -------------------------------------------------------- | -------------------------------------------------------- |
> > | Standard                | **85.32$\pm$0.09**/80.71$\pm$0.17/77.88$\pm$0.13 | 83.19$\pm$0.05/78.69$\pm$0.13/75.34$\pm$0.26             | 82.06$\pm$0.34/77.12$\pm$0.34/73.10$\pm$0.82             |
> > | Reweight-T max          | 84.67$\pm$0.24/80.48$\pm$0.34/77.73$\pm$0.39     | 82.75$\pm$0.20/77.75$\pm$0.09/75.43$\pm$0.17             | 81.54$\pm$0.06/76.68$\pm$0.16/74.36$\pm$0.13             |
> > | Reweight-T 97%          | 84.54$\pm$0.12/78.79$\pm$0.10/77.67$\pm$0.29     | 83.08$\pm$0.34/76.00$\pm$0.32/75.95$\pm$0.33             | 81.17$\pm$0.82/73.01$\pm$0.84/73.92$\pm$1.02             |
> > | Reweight-DualT max      | 84.92$\pm$0.15/**80.96$\pm$0.36**/78.49$\pm$0.39 | **83.42$\pm$0.14**/78.89$\pm$0.22/76.64$\pm$0.18             | 82.29$\pm$0.05/78.87$\pm$0.19/75.57$\pm$0.88             |
> > | Reweight-DualT 97%      | 83.87$\pm$0.24/70.59$\pm$0.19/73.90$\pm$0.14     | 77.97$\pm$0.17/59.64$\pm$0.37/65.79$\pm$0.10             | 75.50$\pm$0.11/57.40$\pm$0.18/60.23$\pm$0.18             |
> > | Reweight-Ours           | 84.75$\pm$0.08/80.87$\pm$0.24/**78.84$\pm$0.24** | 83.34$\pm$0.11/**79.54$\pm$0.02**/**77.56$\pm$0.06** | **82.50$\pm$0.03**/**79.70$\pm$0.04**/**77.49$\pm$0.09** |
> >
> > Table 1-1: Comparison with pair-wise label noise.
> >
> >
> > > **Q3：**The evaluation is unrealistic.
> >
> > **A3:**  To further verify our approach in more realistic cases, we introduce one instance-dependent label noise, PMD label noise [6], into the VOC2007 datasets. For PMD label noise,  data near the decision boundary are harder to distinguish and more likely to be mislabeled, which is much realistic. Comparison for classification performance in this case can be seen in Table 1-2.  The results show our approach can also achieve better performance.
> >
> > | mAP(%)/ OF1(%) / CF1(%) | PMD-Type-I                                           | PMD-Type-II                                          | PMD-Type-III                                     |
> > | ----------------------- | ---------------------------------------------------- | ---------------------------------------------------- | ------------------------------------------------ |
> > | Standard                | 78.03$\pm$0.42/57.55$\pm$2.64/51.95$\pm$2.43         | **82.98$\pm$0.36**/77.07$\pm$0.10/74.07$\pm$0.60     | **82.09$\pm$0.69**/75.93$\pm$0.38/72.65$\pm$0.79 |
> > | Reweight-T max          | 77.74$\pm$0.69/63.58$\pm$3.06/58.70$\pm$0.95         | 82.60$\pm$0.30/77.42$\pm$0.57/74.34$\pm$0.57         | 81.92$\pm$0.59/76.33$\pm$0.92/73.15$\pm$0.77     |
> > | Reweight-T 97%          | 78.04$\pm$0.73/72.00$\pm$1.04/**72.34$\pm$0.41**     | 82.32$\pm$0.59/77.07$\pm$0.47/75.83$\pm$0.21         | 81.88$\pm$0.54/76.63$\pm$0.32/**75.27$\pm$0.34** |
> > | Reweight-DualT max      | 78.20$\pm$0.77/69.30$\pm$3.85/65.35$\pm$2.36         | 82.79$\pm$0.27/77.81$\pm$0.57/74.90$\pm$0.65         | 81.71$\pm$1.04/76.77$\pm$0.78/73.86$\pm$0.67     |
> > | Reweight-DualT 97%      | 75.46$\pm$0.62/60.17$\pm$1.18/62.28$\pm$1.36         | 80.02$\pm$0.22/67.20$\pm$0.32/71.22$\pm$1.00         | 80.39$\pm$0.28/70.03$\pm$2.10/71.50$\pm$1.02     |
> > | Reweight-Ours           | **78.52$\pm$0.65**/**73.16$\pm$2.55**/72.17$\pm$1.64 | 82.50$\pm$0.29/**78.02$\pm$0.37**/**76.22$\pm$0.45** | 81.85$\pm$0.41/**77.26$\pm$0.63**/74.98$\pm$0.62 |
> >
> > Table 1-2: Comparison with PMD label noise (%). The definition of Type-I, Type-II, and Type-III can be found in the original paper [6].
> >
> >
> >
> > > **Q4:** There is no detailed mention of how to resolve the class-imbalance problem.
> >
> > **A4:** The class-imbalance problem we refer to is the positive-negative class imbalance, which makes it difficult for the networks to accurately estimate the noisy posterior probability, and most of transition matrix estimation methods heavily depend on the accurate fitting of the noisy posterior probability.   Since our estimator utilizes label correlations to perform transition matrix estimation, which does not need to accurately estimate the noisy posterior probability,  it naturally avoids this problem in the transition matrix estimation.
> >
> >
> >
> > [6] Learning with Feature-Dependent Label Noise : A Progressive Approach. ICLR 2021.
> >
> > [7] CCMN: A General Framework for Learning with Class-Conditional Multi-Label Noise. TPAMI 2022.
> >
> > [8] Multi-Label Learning with Missing Labels. ICPR 2014.
> >
> > [9] Partial Multi-Label Learning. AAAI 2018.
> >
> > [10] Co-teaching: Robust Training of Deep Neural Networks with Extremely Noisy Labels.  NeurIPS 2018.

---

> > > ### Author Response · Authors · 2022-08-02
> > > **Response to Reviewer hyUM Part (3)**
> > >
> > >
> > > > **Q5:** The mAP score is low on MS-COCO dataset.
> > >
> > > **A5:** The results of some methods under (0, 0) setup on MS-COCO dataset are shown in Table 1-3. CSRA  and AGCN use the default hyperparameters.  In order to complete repeated experiments of different methods under various cases with our limited computing resources,  for MS-COCO datasets, we adopted images with 224 x 224 resolution, and use Adam optimizer with 5e-5 leaning rate and 128 batch size for 30 epochs, which will converge faster, and may have a drop of best mAP score compared with the optimal setting.  In addition, a littile difference is that we use 10% of the training set as the validation set to perform model selection, while some results in the related work are obtained using all training dataset for learning. And according to the previous work [11],  the ResNet-101 using ImageNet pretraining and 224x224 resolution achieves 75.2% mAP,  which is about 2.1% higher than our Standard baseline with ResNet-50. We think this will not affect the effectiveness of the results.
> > >
> > >
> > > |         | Standard |  GCE  |  CDR  | CSRA  |   AGCN    | **Reweight-T max** | Reweight-T 97% | Reweight-DualT max | Reweight-DualT 97% | **Reweight-Ours** |
> > > | ------- | :------: | :---: | :---: | :---: | :-------: | :----------------: | :------------: | :----------------: | ------------------ | ----------------- |
> > > | mAP (%) |  73.14   | 73.37 | 73.16 | 73.85 | **74.61** |       72.99        |     71.27      |       71.59        | 67.97              | 72.74             |
> > > | OF1 (%) |  72.62   | 72.89 | 72.65 | 72.71 | **73.15** |       72.44        |     60.44      |       69.82        | 42.9               | 71.78             |
> > > | CF1 (%) |  68.65   | 68.37 | 68.17 | 69.18 | **69.74** |       68.28        |     58.36      |       68.73        | 44.93              | 69.41             |
> > >
> > > Table 1-3: Comparison on  MS-COCO dataset under (0, 0) setup (%).
> > >
> > >
> > >
> > > > **Q6:**   The paper writing should be improved, particularly the theoretical section.
> > >
> > > **A6:**   Thank you for your suggestion. We have modified the presentation and uploaded the revised version, and especially we detailed the proof of Theorem 3 in Appendix E.   We highly appreciate your insightful comments.
> > >
> > > [11] Learning Spatial Regularization with Image-level Supervisions for Multi-label Image Classification. CVPR 2017.
> > >
> > > [12] Multi-label image recognition with graph convolutional networks. CVPR 2019.
> > >
> > > [13] How does disagreement help generalization against label corruption? ICML 2019.
> > >
> > > [14] Detecting Corrupted Labels Without Training a Model to Predict. ICML 2022.

---

> > > > ### Comment · Reviewer_hyUM · 2022-08-08
> > > > **Response to Authors.**
> > > >
> > > > I appreciate your detailed response to my concerns. This is my opinion on each part:
> > > >
> > > > *Class-independent label noise*:  I understand what it means to use 'class-independent' in this paper, i.e., label 0 and 1 for each class. But, I am not sure if this is class-dependent because the definition of 'class' does not stand for '0' and '1'. It is more appropriate to say 'asymmetric label flipping' in which the flip probability depends on '1' and '0'. I think that the context of using 'class-dependent' should be the same with single-label setups. Although the authors provided an example, where most of the class labels are hardly confused by each other in multi-label learning, there could be large-scale data including many confusing class labels. For example, data with a large number of classes like OpenImages data with 19,957 classes, and data with many confusing attributes in Fashion data like DeepFashion with 1,000 attributes class. Therefore, I don't think the situation will differ a lot compared to the single-label recognition task if we assume the same labeling budget. I recommend the authors clearly state the definition (and the differences) of 'class-dependent' in the setup of the paper so that it will not make ambiguity for readers.
> > > >
> > > > *Results with PMD*:  In addition, the authors additionally perform experiments with PMD label noise in more realistic cases. I agree that this is useful to see the effectiveness of the method. But, it would be better if the authors use MS-COCO data; each instance of VOC only contains less than 2 positive labels on average, which is more like a single-label classification.
> > > >
> > > > **Results with (0,0) setup*: In Table 1-3, except CF1, the proposed method is worse than 'standard', and much worse than AGCN. Could you please discuss the reason (which is a negative/weakness of the proposed methods to use in practice)?

---

> > > > > ### Author Response · Authors · 2022-08-09
> > > > > **Response to Reviewer hyUM Part (4)**
> > > > >
> > > > > Thanks a lot for your feedback. We further address your mentioned concerns as follows.
> > > > >
> > > > > > **Q7:** There could be large-scale data including many confusing class labels. Therefore, I don't think the situation will differ a lot compared to the single-label recognition task if we assume the same labeling budget.
> > > > >
> > > > > **A7:**  **I agree with you that there are many confusing class pairs on the typical large-scale datasets with a large number of classes. While we claim those hardly confused class pairs usually account for the majority of all label pairs for the following reasons:**
> > > > >
> > > > > (1)  Among a large number of classes in the typical large-scale multi-label datasets, e.g. MS-COCO and OpenImages datasets, most label pairs belong to significantly different superclasses, and therefore,  these label pairs are hardly confused. For example,  in MS-COCO datasets,   there are 80 classes, which belong to 10 hardly confused superclasses (outdoor, food, indoor, appliance, sports, person, animal, vehicle, furniture, accessory, electronic, kitchen). In OpenImages dataset, there are 19,957 classes, and also have significantly different superclasses,  such as Toy, Budilding, Medical equipment, Clothing,  Insect and so on (a part of these superclasses can be seen in [17]).
> > > > >
> > > > > (2) Consistent with the above discussion,  the research works about real-world label noise [15,16] show that noisy labels usually flips to some similar class labels in the real-world scene. For example, In CIFAR-100N, which is a re-annotated version of the CIFAR-100 with real-world human annotations,  most classes are more likely to be mislabeled into less than four fine classes[15]. In ANIMAL-10N, the label noise mainly happens between five pairs of confusing animals [16].
> > > > >
> > > > > Hence, on the typical multi-label datasets like MS-COCO and OpenImages datasets, the hardly confused label pairs account for the majority.
> > > > >
> > > > > **In addition, we claim that noisy multi-label learning has various real-world scenes, and some of them can not be seen as the label flips like single-label recognition tasks:**
> > > > >
> > > > > (a) Due to the labeler's confusion, when labeling the music emotion, the annotator may label happy music with "happy" and "existing", which will lead to partial multi-labels.
> > > > >
> > > > > (b)  Due to the complex label correlations, when labeling fashion attributes, people may miss some fabric classes if "Mickey" style label appears, which will lead to missing multi-labels.
> > > > >
> > > > > (c) For saving labeling costs, people manually verified machine-populated labels, which will lead to missing multi-labels, and the large OpenImages dataset is built in this way.
> > > > >
> > > > > (d) For saving labeling costs, annotators may roughly assign each instance a set of candidate labels, which will lead to partial multi-labels.
> > > > >
> > > > > In the real-world noisy multi-label learning,  it even may be the mixture of the label flips like single-label recognition tasks and the above several scenes.  Faced with various complicated real-world multi-label label noise,  it is very hard to be always suitable. Our approach works when the label noise of one class label is only dependent on the label correlations with a few classes,  and nearly independent on the label correlations with most classes.  We think this condition can be nearly satisfied in many real-world scenes, because generally speaking, the multi-label label noises for class j are usually dependent on confusing features for itself, and the majority of classes will not share the same confusing features, which is also verified by the experiments in Table 1-2 and Table 1-5 in the typical datasets.
> > > > >
> > > > > **Besides, thanks for your insightful comments. We will add the discussion about our label noise assumption in the appendices.**
> > > > >
> > > > > > **Q8:** I recommend the authors clearly state the definition (and the differences) of 'class-dependent' in the setup of the paper so that it will not make ambiguity for readers.
> > > > >
> > > > > **A8:** Thanks a lot for your good suggestion.
> > > > >
> > > > > In this paper, the "class-dependent" label noise for class label $Y_j$ means that the flip probabilities are dependent on class label $Y^j=0$ or $Y^j=1$. The differences between this definition and the "class-dependent" label noise in the single-label cases are as follows:
> > > > >
> > > > >  (1) The "class-dependent" label noise in the single-label cases represent the flip probability from class $i$ to class $j$ ($i$ and $j$ are two different classes), while in this paper, the "class-dependent" label noise for class label $Y_j$ is only dependent on $Y^j=0$ or $Y^j=1$, which is independent on another class $Y^i$ , i.e. $P\left(\bar{Y}^{j}| Y^{j}, Y^{i}\right) = P\left(\bar{Y}^{j}| Y^{j} \right)$.
> > > > >
> > > > > [14] Learning with Noisy Labels Revisited: A Study Using Real-World Human Annotations. ICLR 2022.
> > > > >
> > > > > [15] SELFIE: Refurbishing Unclean Samples for Robust Deep Learning. ICML 2019.
> > > > >
> > > > > [16] https://storage.googleapis.com/openimages/2018_04/bbox_labels_600_hierarchy_visualizer/circle.html

---

> > > > > > ### Comment · Reviewer_hyUM · 2022-08-09
> > > > > > **Response.**
> > > > > >
> > > > > > Thanks for the detailed explanation about possible label noise with multi-labels. I agree that hardly confused class pairs usually account for the majority of all label pairs, but also this is not all the case.
> > > > > >
> > > > > > Modeling realistic label noise in a multi-label setup is very challenging. But, I give your study a high value in that your assumption/approach is a possible way as early work on this topic.
> > > > > >
> > > > > > While reading your paper, I had some time to think about what kind of label noise should be considered in the multi-label setup. This was a valuable time to discuss with you and will be of great help to ready my future research.
> > > > > >
> > > > > > I appreciate the authors' active rebuttal and they provided ample opinion about my concerns. My concerns are not completely resolved but this is attributed to a very complex definition of label noise in a multi-label setup. Therefore, I agree with most of the opinions of the authors and, decide to increase my score to '5'.
> > > > > >
> > > > > > Best,

---

> > > > > > > ### Author Response · Authors · 2022-08-09
> > > > > > > **Comment to Reviewer hyUM**
> > > > > > >
> > > > > > > Thanks very much for your careful and insightful review!  We also think these discussions with you are very valuable. We promise to add our discussion and the complete experiments you mentioned in the revised version.

---

> > > > > ### Author Response · Authors · 2022-08-09
> > > > > **Response to Reviewer hyUM Part (5)**
> > > > >
> > > > > (2) The "class-dependent" label noise in the single-label cases can be modeled by  a $C \times C$ transition matrix, bridging the transition  from clean single label to noisy single label,   while in this paper, the "class-dependent" label noise for class label $Y_j$ can be modeled by a $2 \times 2$ transition matrix, bridging the transition from clean label $Y^j$ to noisy label $\bar{Y}^{j}$.
> > > > >
> > > > > We will clearly state the definition and the differences in the revised version.
> > > > >
> > > > >
> > > > >
> > > > > > **Q9:** it would be better if the authors use MS-COCO data under PMD label noise.
> > > > >
> > > > > **A9:** Following your good suggestion, we introduce PMD label noise [6], into the MS-COCO datasets.  Comparison for classification performance can be seen in Table 1-5.  The results verify the effectiveness of our approach in a very realistic scene.
> > > > >
> > > > >
> > > > >
> > > > > | mAP(%)/ OF1(%) / CF1(%) | PMD-Type-I                    | PMD-Type-II               | PMD-Type-III              |
> > > > > | ----------------------- | ----------------------------- | ------------------------- | ------------------------- |
> > > > > | Standard                | 61.04/44.68/39.45             | 64.79/61.47/54.57         | **64.29**/59.37/52.45     |
> > > > > | Reweight-T max          | 60.49/57.84/49.68             | **64.97**/62.37/56.73     | 64.25/59.84/53.24         |
> > > > > | Reweight-T 97%          | 59.41/52.27/50.45             | 63.31/52.73/52.09         | 62.39/52.60/51.90         |
> > > > > | Reweight-DualT max      | 60.51/61.30/60.22             | 62.55/58.64/59.10         | 62.27/58.37/57.96         |
> > > > > | Reweight-DualT 97%      | 53.60/32.18/33.27             | 59.86/33.89/38.22         | 58.35/34.56/38.06         |
> > > > > | Reweight-Ours           | **61.84**/**63.96**/**60.93** | 64.78/**65.14**/**61.05** | 64.19/**63.27**/**60.20** |
> > > > >
> > > > > Table 1-5: Comparison with PMD label noise on MS-COCO dataset (%).
> > > > >
> > > > >
> > > > >
> > > > > > **Q10:** Under (0,0) setup, why is the proposed method worse than 'standard', and much worse than AGCN on mAP and OF1 metrics.
> > > > >
> > > > > **A10:**  (1) Theoretically, when the noise transition matrices are accurately estimated under (0,0) setup, which will be identity matrices, the Reweight method will degenerate into the form of the Standard method [9]. While when the estimation of transition matrices has an error, it will make the empirical risk biased to the ideal expected risk with clean data.  Hence,  the estimation error of transition matrices is the main reason why the Reweight method with estimated matrices, including our proposed method is worse than the Standard method.
> > > > >
> > > > > (2) As shown in Table 1-6, the unbiased Reweight-Ours(tau=0.0) is still a little worse than the Standard method on mAP and OF1 scores.  It may be because we use the loss on validation set, rather than the mAP scores on validation set to select the model after warmup standard training in Reweight methods[2], since we found the loss on validation set is more sutibale to select model for importance reweighting under label noise cases.
> > > > >
> > > > > (3)  To reduce the negative effect of estimation error, we suggest to use another risk-consistent algorithm, T-reversion [1],  which jointly tunes the transition matrices and classifier with Reweight technique. As shown in Table 1-6,  Reversion-Ours(tau=0.5) method can achieve a similar performance compared with the unbiased Reweight-Ours(tau=0.0).
> > > > >
> > > > > (4)  The main reason of the gap between the performances of the proposed method and AGCN  is the well-designed network of AGCN.  Appendix L has shown the risk-consistent methods can also help AGCN perform better under label noise, and here we also test Reweight-AGCN-Ours under (0,0) setup. As shown in Table 1-6,   Reweight-AGCN-Ours  seems be more robust to the estimation error, the mAP of AGCN-R-Ours(tau=0.5) is just about 0.05 lower than AGCN under (0,0) setup.
> > > > >
> > > > >
> > > > >
> > > > > |                    | Standard | Reweight-Ours(tau=0.0) | Reweight-Ours(tau=0.5) | Reversion-Ours(tau=0.5) | AGCN  | AGCN-R-Ours(tau=0.0) | AGCN-R-Ours(tau=0.5) |
> > > > > | ------------------ | :------: | :--------------------: | :--------------------: | :---------------------: | :---: | :------------------: | -------------------- |
> > > > > | mAP (%)            |  73.14   |         72.98          |         72.74          |          72.99          | 74.61 |        74.47         | 74.54                |
> > > > > | OF1 (%)            |  72.62   |         72.19          |         71.78          |          72.08          | 73.15 |        73.21         | 72.71                |
> > > > > | CF1 (%)            |  68.65   |         68.33          |         69.41          |          69.11          | 69.74 |        69.37         | 70.41                |
> > > > > | Estimation   Error |    /     |          0.00          |         13.94          |          11.20          |   /   |         0.00         | 11.43                |
> > > > >
> > > > > Table 1-6: Comparison with different methods on MS-COCO dataset under (0, 0) setup (%).

---

> > > ### Author Response · Authors · 2022-08-02
> > > **Comparison for estimating transition matrices under pair-wise label noise and PMD label noise.**
> > >
> > > We further represent the estimation error between the estimated transition matrices and $P(\bar{Y}^j|Y^j)$ to illustrate the effectiveness of our estimator. As shown in Table 1-4, the proposed estimation method leads to the smallest or second smallest estimator errors across various cases.
> > >
> > > | Estimation Error    |   Pair-wise 10%   |   Pair-wise 15%   |   Pair-wise 20%   |    PMD-Type-I     |    PMD-Type-II    |   PMD-Type-III    |
> > > | ------------------- | :---------------: | :---------------: | :---------------: | :---------------: | :---------------: | :---------------: |
> > > | T-estimator max     |   1.99$\pm$0.02   |   2.97$\pm$0.01   |   3.95$\pm$0.01   |   8.70$\pm$0.13   |   5.66$\pm$0.02   |   6.13$\pm$0.02   |
> > > | T-estimator 97%     |   5.22$\pm$0.02   |   5.53$\pm$0.10   |   5.08$\pm$0.09   | `3.45`$\pm$`0.13` |   4.01$\pm$0.16   | `4.02`$\pm$`0.03` |
> > > | DualT-estimator max | **1.06$\pm$0.03** | **1.36$\pm$0.09** | `1.62`$\pm$`0.06` |   4.31$\pm$0.42   | `3.52`$\pm$`0.07` |   4.33$\pm$0.04   |
> > > | DualT-estimator 97% |  14.49$\pm$0.02   |  14.13$\pm$0.05   |  13.47$\pm$0.04   |   9.35$\pm$0.01   |  10.78$\pm$0.01   |  10.72$\pm$0.01   |
> > > | Our estimator       | `1.82`$\pm$`0.05` | `2.19`$\pm$`0.03` | **1.55$\pm$0.04** | **1.29$\pm$0.07** | **1.71$\pm$0.16** | **1.96$\pm$0.20** |
> > >
> > > Table 1-4: Comparison with the estimation error between the estimated transition matrix  and $P(\bar{Y}^j|Y^j)$.

---

> ### Author Response · Authors · 2022-08-06
> **Comment to Reviewer hyUM**
>
> Thanks a lot for your efforts in reviewing this paper. We tried our best to address your mentioned concerns. Are there unclear explanations? We could further clarify them.

---

> ### Author Response · Authors · 2022-08-07
> **Comment to Reviewer hyUM**
>
> Thank you for reviewing this paper. There are less than three days left until the end of the author-reviewer discussion. We hope you can give some feedback on our response so that we can further explain it if something is unclear.

---

### Meta-Review · Area_Chair_2LQB · 2022-08-28

**Recommendation:** Accept
**Confidence:** Certain

**Metareview:**

Estimating the noisy transition matrix for handling noisy labels with multi-labels.  Good experimental work illustrating estimating transition matrices.  reviewers liked theory and the writeup.   Paper has had improved citations and writing.

There was some discussion about the assumptions.  Nuances of this should be addressed in the revised paper, for instance your comments about class imbalance.

Regarding reviewer KMFh's Q5:  Note retrieval metrics (e.g., R@K) have been widely used in multi-label classification, although versions of F1 are probably more common.  They give alternative looks at the errors.

Regarding Reviewer n5cR's weakness 2: would be nice to do summary plots and/or win/loss tables and put some of the big tables in appendices.



**Award:**

No

---

### Decision · Program_Chairs · 2022-09-14

Accept